# Beyond Slater's Condition in Online CMDPs with Stochastic and Adversarial Constraints

## Abstract

We study *online episodic Constrained Markov Decision Processes* (CMDPs) under both stochastic and adversarial constraints. We provide a novel algorithm whose guarantees greatly improve those of the state-of-the-art best-of-both-worlds algorithm introduced by Stradi et al. (2025c). In the stochastic regime, *i.e.*, when the constraints are sampled from fixed but unknown distributions, our method achieves $\widetilde{\mathcal{O}}(\sqrt{T})$ regret and constraint violation without relying on Slater's condition, thereby handling settings where no strictly feasible solution exists. Moreover, we provide guarantees on the stronger notion of *positive* constraint violation, which does not allow to recover from large violation in the early episodes by playing strictly safe policies. In the adversarial regime, *i.e.*, when the constraints may change arbitrarily between episodes, our algorithm ensures sublinear constraint violation without Slater's condition, and achieves sublinear $\alpha$-regret with respect to the *unconstrained* optimum, where $\alpha$ is a suitably defined multiplicative approximation factor. We further validate our results through synthetic experiments, showing the practical effectiveness of our algorithm.

## 1 Introduction

Reinforcement Learning (RL) (Sutton & Barto, 2018) provides a general framework for sequential decision-making, where an agent learns to act optimally by interacting with an environment modeled as a Markov Decision Process (MDP) (Puterman, 2014). While RL has achieved remarkable success in numerous applications, real-world decision-making problems often involve *safety and resource constraints* that must be respected at every step, leading to the study of *Constrained Markov Decision Processes* (CMDPs) (Altman, 1999). CMDPs have been widely employed in safety-critical domains such as autonomous driving (Isele et al., 2018; Wen et al., 2020), online bidding and advertising (Gummadi et al., 2012; Wu et al., 2018; He et al., 2021), and recommendation systems (Singh et al., 2020), where constraint satisfaction is as crucial as optimizing cumulative reward.

CMDPs have been significantly studied within the framework of *online learning* (Cesa-Bianchi & Lugosi, 2006), where a learner interacts with an environment in a sequential manner and aims to minimize its *regret*, defined as the difference between the reward attained by the best fixed policy and the learner's cumulative reward. An algorithm is considered successful if it achieves *sublinear regret*, meaning that the average regret per round vanishes as the time horizon $T$ grows. Online CMDPs extend this setting by incorporating constraints on the learner's behavior, making them a constrained counterpart of classical online learning problems. These algorithms are typically studied under two main assumptions about the environment: in the *stochastic* setting, rewards (losses) and constraint functions are drawn i.i.d. from an unknown but fixed distribution, while in the *adversarial* setting they can be chosen arbitrarily by an adversary, potentially depending on past actions.

In the stochastic setting, several works provide algorithms for CMDPs that achieve sublinear regret and sublinear constraint violation under various assumptions (e.g., (Efroni et al., 2020; Zheng & Ratliff, 2020; Stradi et al., 2025b)). Adversarial settings, however, are inherently more challenging. In particular, Mannor et al. (2009) show that even in the simple single-state case, when constraints are adversarially chosen, it is *impossible* to guarantee both sublinear regret and sublinear cumulative constraint violation with respect to a fixed policy that satisfies the constraints in hindsight. As a result, most advances in adversarial CMDPs focus on CMDPs with adversarial rewards and stochastic constraints (Qiu et al., 2020; Stradi et al., 2025a). The only exceptions so far are two re-

cent works (Stradi et al., 2024a; 2025c), which introduce the first *best-of-both-worlds* algorithms for episodic CMDPs that can also handle adversarially chosen constraints. In stochastic settings, these methods achieve $\widetilde{\mathcal{O}}(1/\rho^2\sqrt{T})$ regret and constraint violation under a Slater-like feasibility condition, where $\rho$ is a suitably defined Slater's parameter, and $\widetilde{\mathcal{O}}(T^{3/4})$ guarantees without such a condition. Differently, in the adversarial regime (adversarial constraints), they attain sublinear violation and sublinear $\alpha$-regret, with $\alpha = \mathcal{O}(\rho/1+\rho)$, that is, sublinear regret with respect to a fraction of the *constrained* optimum. In the adversarial setting, the algorithms require the Slater's like condition.

Due to space constraints, we refer to Appendix A for a comprehensive discussion on related works.

## 1.1 ORIGINAL CONTRIBUTIONS

We study online episodic CMDPs where the constraints may be either stochastic or adversarial. We propose a novel algorithm that greatly improves the state-of-the-art best-of-both-worlds results provided in (Stradi et al., 2025c). Specifically, in the stochastic setting, our algorithm attains $\widetilde{\mathcal{O}}(\sqrt{T})$ regret $R_T$ and violation $V_T$ without Slater's condition, *i.e.*, even when a strictly feasible solution does not exist. Furthermore, our algorithm attains $\widetilde{\mathcal{O}}(\sqrt{T})$ positive constraint violation $\mathcal{V}_T$, which does not allow for cancellations between episodes.

Table 1: Comparison between our algorithm and the state-of-the-art best-of-both-worlds results.

| | Stradi et al. (2025c) | Algorithm 1 |
|---|---|---|
| $R_T$ Stoc. Constraints | $\widetilde{\mathcal{O}}\left(\min\left\{\frac{1}{\rho^2}\sqrt{T}, T^{\frac{3}{4}}\right\}\right)$ | $\widetilde{\mathcal{O}}(\sqrt{T})$ |
| $V_T$ Stoc. Constraints | $\widetilde{\mathcal{O}}\left(\min\left\{\frac{1}{\rho^2}\sqrt{T}, T^{\frac{3}{4}}\right\}\right)$ | $\widetilde{\mathcal{O}}(\sqrt{T})$ |
| $\mathcal{V}_T$ Stoc. Constraints | ✗ | $\widetilde{\mathcal{O}}(\sqrt{T})$ |
| $\alpha$-$R_T$ Adv. Constraints | $\widetilde{\mathcal{O}}\left(\frac{1}{\rho^2}\sqrt{T}\right)$ | $\widetilde{\mathcal{O}}(\sqrt{T})$ |
| $V_T$ Adv. Constraints | $\widetilde{\mathcal{O}}\left(\frac{1}{\rho^2}\sqrt{T}\right)$ | $\widetilde{\mathcal{O}}(\sqrt{T})$ |

This metric is indeed stronger than the standard constraint violation since it does not allow to recover from large violation in the early episodes by playing strictly safe policies. In the adversarial setting, our algorithm attains sublinear violation without Slater's condition. Furthermore, by employing a slightly stronger notion of Slater's parameter, our algorithm attains sublinear $\alpha$-regret with respect to the *unconstrained* optimum, instead of the constrained one. Finally, we complement our analysis with synthetic experiments that empirically validate our results.

Our contributions are summarized in Table 1.

## 2 PRELIMINARIES

In this section, we provide notation and definitions needed in the rest of the paper.

### 2.1 CONSTRAINED MARKOV DECISION PROCESSES

We study CMDPs (Altman, 1999) defined as tuples $(X, A, P, \{r_t\}_{t=1}^T, \{G_t\}_{t=1}^T)$. Specifically, $T$ is a number of episodes of the learning dynamic, with $t \in [T]$ denoting a specific episode.[1] $X$ and $A$ are finite state and action spaces, respectively. $P : X \times A \to \Delta_{|X|}$ is the transition function,[2] where, for ease of notation, we denote by $P(x'|x, a)$ the probability of going from state $x \in X$ to $x' \in X$ by taking action $a \in A$.[3] $\{r_t\}_{t=1}^T$ is a sequence of vectors describing the rewards at each episode $t \in [T]$, namely $r_t \in [0, 1]^{|X \times A|}$. We refer to the reward of a specific state-action pair $x \in X, a \in A$ for an episode $t \in [T]$ as $r_t(x, a)$. Rewards are *adversarial*, namely, no statistical

---

[1]We denote with $[a, \ldots, b]$ the set of all consecutive integers from $a$ to $b$, while $[b] = [1, \ldots, b]$.

[2]We denote as $\Delta_n$ the $n-1$ dimensional simplex.

[3]W.l.o.g., in this work we consider *loop-free* CMDPs. Formally, this means that $X$ is partitioned into $L$ layers $X_0, \ldots, X_L$ such that the first and the last layers are singletons, *i.e.*, $X_0 = \{x_0\}$ and $X_L = \{x_L\}$, and that $P(x'|x, a) > 0$ only if $x' \in X_{k+1}$ and $x \in X_k$ for some $k \in [0, \ldots, L-1]$. Notice that any episodic CMDP with horizon $L$ that is *not* loop-free can be cast into a loop-free one by suitably duplicating the state space $L$ times, *i.e.*, a state $x$ is mapped to a set of new states $(x, k)$, where $k \in [0, \ldots, L]$.

assumptions are made. $\{G_t\}_{t=1}^T$ is a sequence of constraint matrices describing the $m$ *constraint costs* at each episode $t \in [T]$, namely $G_t \in [-1, 1]^{|X \times A| \times m}$, where non-strictly positive cost values stand for satisfaction of the constraints. For $i \in [m]$, we refer to the cost of the $i$-th constraint for a specific state-action pair $x \in X, a \in A$ at episode $t \in [T]$ as $g_{t,i}(x, a)$. Constraint costs may be *stochastic* (we will refer to this case as stochastic setting), in that case $G_t$ is a random variable distributed according to a probability distribution $\mathcal{G}$ for every $t \in [T]$, or chosen by an *adversary* (we will refer to this case as adversarial setting).

The learner chooses a *policy* $\pi : X \to \Delta_{|A|}$ at each episode, defining a probability distribution over actions at each state. For ease of notation, we denote by $\pi(\cdot|x)$ the probability distribution at $x \in X$, with $\pi(a|x)$ denoting the probability of action $a \in A$.

Protocol 1 provides the complete interaction between the learner and the environment.

Given a transition function $P$ and a policy $\pi$, the *occupancy measure* $q^{P,\pi} \in [0, 1]^{|X \times A \times X|}$ induced by $P$ and $\pi$ (Rosenberg & Mansour, 2019a) is such that, for every $x \in X_k$, $a \in A$, and $x' \in X_{k+1}$ with $k \in [0, \ldots, L-1]$, it holds:

**Protocol 1** Learner-Environment Interaction

1: **for** $t = 1, \ldots, T$ **do**
2:     $r_t$ is chosen *adversarially*
3:     $G_t$ is chosen either *stochastically* or *adversarially*
4:     The learner chooses a policy $\pi_t$
5:     The state is initialized to $x_0$
6:     **for** $k = 0, \ldots, L-1$ **do**
7:         The learner plays $a_k \sim \pi_t(\cdot|x_k)$
8:         The learner observes $r_t(x_k, a_k), g_{t,i}(x_k, a_k) \; \forall i \in [m]$
9:         The environment evolves to $x_{k+1} \sim P(\cdot|x_k, a_k)$
10:        The learner observes $x_{k+1}$
11:     **end for**
12: **end for**

$$q^{P,\pi}(x, a, x') = \mathbb{P}\{x_k = x, a_k = a, x_{k+1} = x' \mid P, \pi\}.$$

Moreover, we let $q^{P,\pi}(x, a) = \sum_{x' \in X_{k+1}} q^{P,\pi}(x, a, x')$ and $q^{P,\pi}(x) = \sum_{a \in A} q^{P,\pi}(x, a)$. An occupancy measures $q \in [0, 1]^{|X \times A \times X|}$ is *valid* if and only if the following three conditions hold:

*(i)* $\sum_{x \in X_k} \sum_{a \in A} \sum_{x' \in X_{k+1}} q(x, a, x') = 1 \;\; \forall k \in [0, \ldots, L-1]$

*(ii)* $\sum_{a \in A} \sum_{x' \in X_{k+1}} q(x, a, x') = \sum_{x' \in X_{k-1}} \sum_{a \in A} q(x', a, x) \;\; \forall k \in [1, \ldots, L-1], \forall x \in X_k$

*(iii)* $P^q = P$,

where $P$ is the transition function of the MDP and $P^q$ is the one induced by $q$ (see below).

Notice that any valid occupancy measure $q$ induces a transition function $P^q$ and a policy $\pi^q$, which are defined as $P^q(x'|x, a) = q(x, a, x')/q(x, a), \pi^q(a|x) = q(x, a)/q(x)$.

**Remark 2.1** (On the stochastic rewards setting). *As pointed out in Protocol 1, we focus exclusively on the adversarial reward setting, unlike for the constraints, where both stochastic and adversarial scenarios are analyzed. This is because the stochastic reward setting follows directly from the adversarial reward one by a straightforward application of the Azuma–Hoeffding inequality.*

## 2.2 BASELINE FOR THE STOCHASTIC SETTING

We define the safe optimum for the stochastic constraints setting as follows:

$$\text{OPT}_{\overline{G}} := \begin{cases} \max_{q \in \Delta(M)} & \frac{1}{T} \sum_{t=1}^T r_t^\top q \\ \text{s.t.} & \overline{G}^\top q \leq \underline{0}, \end{cases} \tag{1}$$

where $q \in [0, 1]^{|X \times A|}$ is the occupancy measure vector, $\Delta(M)$ is the set of valid occupancy measures, and $\overline{G}$ is the expected value of $\mathcal{G}$. Thus, we introduce the notion of *cumulative regret* as:

$$R_T := T \cdot \text{OPT}_{\overline{G}} - \sum_{t=1}^T r_t^\top q^{P,\pi_t}.$$

We refer to an optimal safe occupancy measure (*i.e.*, a feasible one achieving value $\text{OPT}_{\overline{G}}$) as $q^*$. Thus, the regret reduces to $R_T = \sum_{t=1}^T r_t^\top q^* - \sum_{t=1}^T r_t^\top q^{P,\pi_t}$.

### 2.3 BASELINE FOR THE ADVERSARIAL SETTING

In the adversarial case, we define the *cumulative $\alpha$-regret* as follows:

$$\alpha\text{-}R_T = \alpha T \cdot \text{OPT} - \sum_{t=1}^{T} r_t^\top q^{P,\pi_t},$$

where the unconstrained optimal value is defined as $\text{OPT} := \max_{q \in \Delta(M)} \frac{1}{T} \sum_{t=1}^{T} r_t^\top q$. In order to quantify $\alpha$, we introduce a problem-specific parameter $\rho \in [0,1]$, which is defined as $\rho := \max_{q \in \Delta(M)} \min_{(x,a) \in \mathcal{Q}(q)} \min_{t \in [T]} \min_{i \in [m]} -g_{t,i}(x,a)$, where $\mathcal{Q}(q) := \{(x,a) \in X \times A : q(x,a) > 0\}$. Thus, we define $\alpha := \rho/1+\rho$. We denote the occupancy measure leading to the value of $\rho$ as $q^\diamond$. Intuitively, $\rho$ represents the "margin" by which the "most feasible" strictly feasible occupancy satisfies the constraints, in the "worst" state-action pair. Our definition of $\rho$ is slightly stronger than the one employed in (Stradi et al., 2025c), where the problem-specific parameter is not computed with respect to the "worst" state-action pair. Nonetheless, we underline that the baseline employed for the adversarial setting is the unconstrained optimum, while Stradi et al. (2025c) provide no-$\alpha$ regret guarantees with respect to the *constrained* optimum, only.

### 2.4 CONSTRAINT VIOLATION

In order to deal with the problem of satisfying the constraints, we define the cumulative constraint violation up to episode $T$. We underline that this metric is equivalent for both the stochastic and the adversarial setting. Specifically, the *cumulative constraint violation* is defined as:

$$V_T := \max_{i \in [m]} \sum_{t=1}^{T} g_{t,i}^\top q^{P,\pi_t}.$$

Additionally, for the stochastic setting, we study the expected *positive cumulative constraint violation*, which is defined as:

$$\mathcal{V}_T := \max_{i \in [m]} \sum_{t=1}^{T} \left[ \bar{g}_i^\top q^{P,\pi_t} \right]^+,$$

where $[\cdot]^+ := \max\{\cdot, 0\}$ and $\bar{g}_i$ is the $i$-th component of $\overline{G}$. Intuitively, the positive violation metric prevents compensation across episodes; in other words, it is not possible to play largely safe policies in order to recover from the large violation attained in early episodes. For the sake of notation, we will refer to $q^{P,\pi_t}$ by using $q_t$, thus omitting the dependence on $P$ and $\pi_t$.

In this work, we propose an algorithm capable of attaining sublinear regret and (positive) violation guarantees in the stochastic setting—namely, $R_T = o(T), V_T = o(T), \mathcal{V}_T = o(T)$—, while getting sublinear $\alpha$-regret and violation in the adversarial case—namely, $\alpha\text{-}R_T = o(T), V_T = o(T)$.

## 3 ALGORITHM

In this section, we describe the key components of *Weighted Constrained Optimistic Policy Search* (`WC-OPS`, for short), which is the main algorithmic contribution of this paper. In Algorithm 1, we provide the pseudocode of `WC-OPS`.

### 3.1 INITIALIZATION AND LOSS ESTIMATION

Algorithm 1 receives as input the time horizon $T$, the set of states $X$, the set of actions $A$, the learning rate $\eta$, the implicit exploration factor $\gamma$, and the confidence $\delta \in (0,1)$. The occupancy measure $\widehat{q}_1$ is initialized uniformly over all tuples $(x_k, a, x_{k+1}) \in X_k \times A \times X_{k+1}$ for each layer $k \in [0, \ldots, L-1]$. The transition function confidence set $\mathcal{P}_1$ is initialized as the set of all the possible transition functions. The counters $N_t(x,a)$ and $M_t(x'|x,a)$, which are respectively defined as $N_t(x,a) = \sum_{\tau=1}^{t-1} \mathbb{I}_\tau\{x,a\}$ for all $(x,a) \in X \times A$, $M_t(x'|x,a) = \sum_{\tau=1}^{t-1} \mathbb{I}_t\{x,a,x'\}$ for all $(x,a,x') \in X_k \times A \times X_{k+1}, k \in [0,...,L-1]$, are initialized to 0 (see Line 1). We denote by $\mathbb{I}_t\{x,a\}$ and $\mathbb{I}_t\{x,a,x'\}$ the indicator functions for the state-action(-state) visit at episode $\tau$.

---

**Algorithm 1** Weighted Constrained Optimistic Policy Search (`WC-OPS`)

---

**Require:** $T, X, A, \eta, \gamma, \delta$

1: Initialize occupancy $\widehat{q}_1 \leftarrow \frac{1}{|X_k||A||X_{k+1}|}$, the estimated transitions space $\mathcal{P}_1$ as the set of all the possible transition functions, and counters $N_1(x,a) = M_1(x'|x,a) = 0$ for all $k \in [0, ..., L-1]$ and $(x, a, x') \in X_k \times A \times X_{k+1}$

2: **for** $t \in [T]$ **do**

3:      Play policy $\pi_t \leftarrow \pi^{\widehat{q}_t}$

4:      Observe feedback as in Protocol 1

5:      Set $\ell_t(x,a) \leftarrow 1 - r_t(x,a)\mathbb{I}_t\{x,a\}$ for all $x \in X, a \in A$

6:      Compute $\widehat{\ell}_t(x,a) = \frac{\ell_t(x,a)}{u_t(x,a)+\gamma}\mathbb{I}_t\{x,a\}$

7:      Update counters and compute weights as shown in Equation (2)

8:      Compute $\widehat{g}_{t,i}(x,a) = \sum_{\tau \in \mathcal{T}_{t,x,a}} w_{t,x,a,i}(\tau) g_{\tau,i}(x,a)$ for all $x \in X, a \in A, i \in [m]$

9:      Update confidence set $\mathcal{P}_t$ and bonus $b_t$ as prescribed in Equations (3)–(4)

10:     $\widehat{\Delta}_t(\mathcal{P}_t) \leftarrow \left\{ q \in \Delta(\mathcal{P}_t) : (\widehat{g}_{t,i} - b_t)^\top q \le 0 \ \ \forall i \in [m] \right\}$

11:     Update $\widehat{q}_{t+1} \leftarrow \arg\min_{q \in \widehat{\Delta}_t(\mathcal{P}_t)} \widehat{\ell}_t^\top q + \frac{1}{\eta} B(q||\widehat{q}_t)$

12: **end for**

---

At the beginning of each episode $t$, the algorithm executes the policy $\pi_t$ induced by the occupancy measure $\widehat{q}_t$ computed at the previous episode (Line 3). After selecting the policy, the learner interacts with the environment and receives the feedback (Line 4). The loss vector $\ell_t$ is built from the observed reward vector $r_t$ (Line 5). Then, the algorithm builds a *biased* loss estimator $\widehat{\ell}_t$ for episode $t$, following the optimistic approach originally proposed in (Neu, 2015; Jin et al., 2020). Specifically, given the transition function confidence set $\mathcal{P}_t$—refer to Equation (3) for additional details—, which contains the true transition function with high probability, the algorithm builds an optimistic estimator of $\ell_t$. This is done by employing an upper bound on the occupancy $u_t$, in place of the unknown true occupancy $q_t$, defined as $u_t(x,a) = \max_{P_t \in \mathcal{P}_t} q^{P_t, \pi_t}(x,a)$ for all $(x,a) \in X \times A$. This upper bound represents the maximum probability of visiting $(x,a)$ under any transition function within the set $\mathcal{P}_t$. Thus, the estimator is computed as $\widehat{\ell}_t(x,a) = \frac{\ell_t(x,a)}{u_t(x,a)+\gamma}\mathbb{I}_t\{x,a\}$, where $\gamma$ is the implicit exploration factor given as input (Line 6).

### 3.2 WEIGHTS ESTIMATION

At each episode, the counters are updated given the trajectory observed as feedback, namely, $N_t(x,a)$ and $M_t(x'|x,a)$ are updated by incrementing by 1 the entries of the tuples visited during the current episode. Then, the algorithm sets the weights that will be used to build the constraint estimates (Line 7). Specifically, given a pair $(x,a) \in X \times A$, $i \in [m]$, and $t \in [T-1]$, the weights $w_{t,x,a,i}$ are defined as follows:

$$w_{t,x,a,i}(\tau) := \beta_{\tau,i}(x,a) \prod_{h \in \mathcal{T}_{t-1,x,a}:h>\tau} (1 - \beta_{h,i}(x,a)) \quad \forall \tau \in \mathcal{T}_{t-1,x,a}, \tag{2}$$

where $\mathcal{T}_{t,x,a}$ is the set of episodes where the pair $(x,a)$ has been visited up to episode $t$, that is:

$$\mathcal{T}_{t,x,a} := \{\tau \le t : \mathbb{I}_\tau\{x,a\} = 1\}.$$

Moreover, the constraints learning rates $\beta_{t,i}$ are defined as:

$$\beta_{t,i}(x,a) := \frac{1}{N_t(x,a)}(1 + \Gamma_{t,i}),$$

where $\Gamma_{t,i}$ is an adaptive term that depends on the constraint vectors observed and is defined as:

$$\Gamma_{t,i} := \left[ \sum_{\tau \in [t-1]} \sum_{x,a} g_{\tau,i}(x,a)\mathbb{I}_\tau\{x,a\} - \mathcal{C} \right]_0^{\mathcal{C}},$$

$\mathcal{C} := 21L|X|\sqrt{2t|A|\ln\frac{2mT^2|X||A|}{\delta}}$ and $[\cdot]_a^b := \min(\max(\cdot, a), b)$. Finally, the weights are employed to build the estimates $\widehat{g}_{t,i}$ for each constraint $i \in [m]$ and each $(x,a)$ as the weighted mean

of the values observed during the rounds in $\mathcal{T}_{t,x,a}$ (Line 8). Intuitively, the $\Gamma_t$ parameter allows the learning rates to meet the requirements of both the stochastic and the adversarial setting, as we point out in the following. In order to better understand it, we first introduce the following result.

**Proposition 3.1.** *If $\beta_{t,i}(x,a) = \frac{1}{N_t(x,a)}$ for every $\tau \in \mathcal{T}_{t-1,x,a}$ such that $x_\tau = x, a_\tau = a$, then the following holds:*

$$w_{t,x,a,i}(\tau) = \frac{1}{N_t(x,a)},$$

*and we recover the empirical mean estimator:*

$$\widehat{g}_{t,i}(x,a) = \frac{1}{N_t(x,a)} \sum_{\tau \in \mathcal{T}_{t-1,x,a}} g_{\tau,i}(x,a).$$

Proposition 3.1 simply states that, when $\Gamma_{t,i} = 0$, the weighted approach is equivalent to the empirical mean estimator. Indeed, as we will show in Section 4, this is exactly the case when the constraints are stochastic and the empirical mean estimator is sufficient to estimate the constraints. Differently, in the adversarial case, the learning rate is proportional to the violation attained by the algorithm, thus allowing $\widehat{g}_{t,i}$ to move accordingly to the attained performance.

### 3.3 DECISION SPACE DEFINITION AND OPTIMIZATION UPDATE

Given the constraints estimates, Algorithm 1 has to properly build the decision space at each episode. Indeed, the algorithm has to ensure that such a decision space includes the true transition function and the true constraint functions, with high probability. In order to do that, Algorithm 1 updates its model (Line 9) accordingly.

For the transitions, we follow the approach of Rosenberg & Mansour (2019b). Specifically, the transition function confidence set $\mathcal{P}_t$ is updated as follows:

$$\mathcal{P}_t = \left\{ \widehat{P} : \left| \widehat{P}(x'|x,a) - \bar{P}_t(x'|x,a) \right| \leq \epsilon_t(x'|x,a) \right\}, \tag{3}$$

where the confidence width $\epsilon_t(x'|x,a)$ is defined as:

$$\epsilon_t(x,a) = \sqrt{\frac{2|X_{k(x)+1}| \ln \frac{T|X||A|}{\delta}}{\max 1, N_t(x,a)}}, \quad \forall (x,a) \in X \times A,$$

and the empiric transition $\bar{P}_t$ is defined as:

$$\bar{P}_t(x'|x,a) = \frac{M_t(x'|x,a)}{\max\{1, N_t(x,a)\}} \quad \forall (x,a,x') \in X_k \times A \times X_{k+1}, k \in [0, \ldots, L-1].$$

Given $\mathcal{P}_t$, it is possible to build $\Delta(\mathcal{P}_t)$ as the set of all possible occupancy measures.

For the constraints, we build optimistic bonuses $b_t(x,a)$ that are computed as:

$$b_t(x,a) = \sqrt{\frac{2 \ln \frac{2m|X||A|T}{\delta}}{N_t(x,a)}} \quad \forall (x,a) \in X \times A. \tag{4}$$

At each episode, the algorithm estimates the per-episode decision space $\Delta_t(\mathcal{P}_t)$ taking the intersection between $\Delta(\mathcal{P}_t)$ and the space of optimistically safe occupancy measures such that $(\widehat{g}_{t,i} - b_t)^\top q \leq 0$ for all $i \in [m]$ (Line 10). We underline that the bonus quantity $b_t$ is necessary for the stochastic setting only, that is, when the empirical mean estimation is employed for the constraints. In the adversarial setting, the constraints estimator $\widehat{g}_{t,i}$ is sufficient to attain the desired theoretical guarantees.

Finally, the algorithm employs an online mirror descent (OMD) (Orabona, 2019) update step on the estimated feasible set $\Delta_t(\mathcal{P}_t)$ (Line 11) employing the unnormalized Kullback-Leibler divergence as the Bregman divergence. Formally:

$$B(q\|\widehat{q}_t) = \sum_{x,a,x'} q(x,a,x') \ln \frac{q(x,a,x')}{\widehat{q}_t(x,a,x')} - \sum_{x,a,x'} (q(x,a,x') - \widehat{q}_t(x,a,x')).$$

**Remark 3.2** (Algorithmic comparison with (Stradi et al., 2025c)). *Algorithm 1 employs a completely different approach with respect to the state-of-the-art best-of-both-worlds algorithm for CMDPs presented in (Stradi et al., 2025c). Specifically, Stradi et al. (2025c) propose a primal-dual method, where a primal no-regret algorithm optimizes the Lagrange function of the CMDP, while a dual no-regret algorithm selects the most violated constraint. Our approach is substantially different since we do not make any use of the Lagrangian formulation of the CMDP. Differently, we resort to a "moving" decision space approach, where we employ a no-regret optimization update over a decision space that adaptively follows the constraints estimation. As we show in Section 4, this technique allows us to be particularly effective when the constraints are stochastic. In this case, we have no need of any Slater's like condition, as the constraints are estimated using a UCB-like approach. Crucially, the "moving" decision space still allows us to recover sublinear violation and sublinear $\alpha$-regret in the adversarial setting.*

## 4 THEORETICAL RESULTS

In this section, we prove the theoretical guarantees attained by Algorithm 1. Specifically, we first discuss the stochastic setting. Then, we show the performance of our algorithm when the constraints are adversarial.

### 4.1 STOCHASTIC SETTING

In this section, we focus on the stochastic setting, that is, the constraints are sampled from fixed distributions. The first fundamental result is to show that the bonus terms $b_t(x, a)$ encompass the distance between the constraints estimator and the true constraint function. This is done by means of the following lemma.

**Lemma 4.1.** *Let $\delta \in (0, 1)$. In the stochastic setting, with probability at least $1 - 11\delta$, it holds that:*

$$|\widehat{g}_{t,i}(x, a) - \bar{g}_i(x, a)| \leq b_t(x, a) \quad \forall (x, a) \in X \times A, i \in [m], t \in [T].$$

Intuitively, the result is proved as follows. We proceed by induction. In the first episodes, $\Gamma_{t,i} = 0$ for all $i \in [m]$. Thus, by Proposition 3.1, the constraint estimator is computed as $\widehat{g}_{t,i} = \frac{1}{N_t(x,a)} \sum_{\tau \in \mathcal{T}_{t-1,x,a}} g_{\tau,i}(x, a)$, that is, the sample mean of the observed constraints values. Employing an Hoeffding bound, it is easy to see that Lemma 4.1 holds for those specific episodes. The induction step consists in showing that, assuming $\sum_{\tau \in [t-1]} \sum_{x,a} g_{\tau,i}(x, a)\mathbb{I}_\tau\{x, a\} \leq \mathcal{C}$ at episode $t - 1$, the same holds for the violation observed at $t$, too. This is done by showing that the empirical mean estimator and the bonus term are sufficient to keep the violation small when the constraints are stochastic. Again, since we proved that $\sum_{\tau \in [t]} \sum_{x,a} g_{\tau,i}(x, a)\mathbb{I}_\tau\{x, a\} \leq \mathcal{C}$, we have $\Gamma_{t,i} = 0$, which concludes the proof after a simple application of the Hoeffding inequality.

Given Lemma 4.1, the following corollary immediately holds.

**Corollary 4.2.** *In the stochastic setting, let $\delta \in (0, 1)$ and $\Delta^\star = \{q \in \Delta(M) : \bar{g}_i^\top q \leq 0 \ \forall i \in [m]\}$. Then, with probability at least $1 - 11\delta$, it holds:*

$$\Delta^\star \subseteq \widehat{\Delta}_t(\mathcal{P}_t) \quad \forall t \in [T].$$

Corollary 4.2 simply states that the true safe decision space is included in the per-episode decision space. This is intuitive, since, by Lemma 4.1, subtracting the bonus term to the constraints estimator allows, with high probability, to be optimistic in the constraints definition. A similar reasoning holds for the transitions. We are now ready to show the main result of the section, that is, the final regret and violation bound. This is done in the following theorem.

**Theorem 4.3.** *Let $\delta \in (0, 1)$. In the stochastic setting, Algorithm 1, with $\eta = \gamma = \sqrt{\frac{L \ln(L|X||A|/\delta)}{T|X||A|}}$, guarantees that with probability at least $1 - 30\delta$:*

$$R_T \leq 14L|X|^2 \sqrt{2T|A| \ln\left(\frac{T|X|^2|A|}{\delta}\right)},$$

*and*

$$V_t \leq 61L|X| \sqrt{2t|A| \ln\left(\frac{2mT^2|X||A|}{\delta}\right)}, \quad \forall t \in [T].$$

Theorem 4.3 follows from the following reasoning. As concerns the regret bound, by Corollary 4.2, it holds that the safe optimum is included in the per-episode decision space, with high probability. Thus, following a standard no-regret argument of OMD with implicit exploration shows that Algorithm 1 attains sublinear regret with respect to any occupancy which is included in the algorithm decision space at each episode. Differently, to prove the violation, we proceed by contradiction, that is, we show that the weights definition does not allow the violation to exceed the threshold defined by the bound of Theorem 4.3. We remark that the proof for the violation is equivalent to the one for the adversarial setting, since the definition of $V_t$ is equivalent between the two settings. Indeed, in this case, we do not have to exploit Corollary 4.2, since even when $\Gamma_{t,i} = C$, the violations are not allowed to exceed the aforementioned value.

We conclude the section by providing the positive violation bound attained by Algorithm 1.

**Theorem 4.4.** *Let $\delta \in (0,1)$. In the stochastic setting, Algorithm 1 guarantees with probability at least $1 - 16\delta$:*

$$\mathcal{V}_t \leq 18L|X|\sqrt{2t|A|\ln\frac{2mT|X||A|}{\delta}}, \quad \forall t \in [T].$$

Intuitively, Theorem 4.4 is proved by showing that the positive violation attained by Algorithm 1 is proportional to the bonus $b_t$ term employed in the decision space definition. Showing that the term concentrate at a $1/\sqrt{T}$ rate concludes the proof. We finally remark that the results provided in this section strongly improve the ones provided in (Stradi et al., 2025c) for the stochastic setting, as we highlight in the following. First, Algorithm 1 does not rely on any Slater's like condition to attain the optimal $\widetilde{\mathcal{O}}(\sqrt{T})$ regret and violation bounds. Second, Algorithm 1 attains the optimal rate for the *positive* constraints violation metric.

### 4.2 ADVERSARIAL SETTING

In this section, we focus on the adversarial setting, that is, the constraints are allowed to change arbitrarily over episodes. In such a setting, Mannor et al. (2009) showed the impossibility to attain sublinear regret and violation, simultaneously. Thus, as is standard in the constrained online learning literature (Castiglioni et al., 2022a; Stradi et al., 2025c), we focus on attaining sublinear violation and sublinear $\alpha$-regret. Similarly to the stochastic setting, we show that the per-episode decision space is well defined. This is done by means of the following theorem.

**Theorem 4.5.** *In the adversarial setting, let $\delta \in (0,1)$ and $\Delta^\diamond$ be the interpolation of any point $q \in \Delta(M)$ and $q^\diamond$ and let $\rho' = L \cdot \rho$. Formally,*

$$\Delta^\diamond := \frac{L}{L+\rho'}\{q^\diamond\} + \frac{\rho'}{L+\rho'}\Delta(M).$$

*Then, with probability at least $1 - \delta$, it holds that $\Delta^\diamond \subseteq \widehat{\Delta}_t(\mathcal{P}_t)$ for all $t \in [T]$.*

Intuitively, Theorem 4.5 shows that any $\alpha$-optimum is included in the per-episode decision space, with high probability. The result is proved employing the definition of the weights and the one of the problem specific parameter $\rho$. We remark that the quantity $\frac{\rho}{1+\rho}$ is equivalent to $\frac{\rho'}{L+\rho'}$, by definition.

We conclude providing the final result of the paper.

**Theorem 4.6.** *Let $\delta \in (0,1)$. In the adversarial setting, Algorithm 1, with $\eta = \gamma = \sqrt{\frac{L\ln(L|X||A|/\delta)}{T|X||A|}}$, guarantees that with probability at least $1 - 19\delta$:*

$$\alpha\text{-}R_T \leq 14L|X|^2\sqrt{2T|A|\ln\left(\frac{T|X|^2|A|}{\delta}\right)}$$

*and*

$$V_t \leq 61L|X|\sqrt{2t|A|\ln\left(\frac{2mT^2|X||A|}{\delta}\right)}$$

*for all $t \in [T]$, where $\alpha = \frac{\rho}{1+\rho}$.*

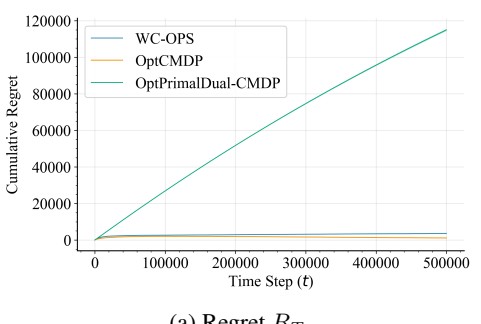
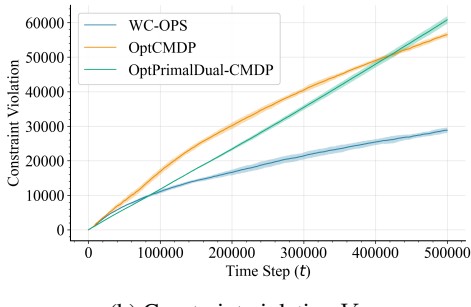

(a) Regret $R_T$         (b) Constraint violation $V_T$

Figure 1: Experimental evaluation of Algorithm 1 (WC-OPS).

Theorem 4.6 is proved employing a similar approach to the one of Theorem 4.3. Specifically, the $\alpha$-regret follows from noticing that, by Theorem 4.5, the $\alpha$-optimum is contained in the per-episode decision space. Thus, employing the OMD with implicit exploration theoretical guarantees gives the result. For the violation, the analysis is equivalent to the one of Theorem 4.3. Comparing the theoretical guarantees of Algorithm 1 and the ones provided in (Stradi et al., 2025c), the following remarks are in order. First, the violation bound provided by Algorithm 1 neither relies on the Slater's condition nor has any dependence on the Slater's parameter. Second, our $\alpha$-regret is computed with respect to the *unconstrained* optimum, rather than the constrained one. Moreover, our bound does not rely on the Slater's parameter of the problem, whereas only the definition of $\alpha$-regret does.

## 5 EXPERIMENTAL EVALUATION

In this section, we evaluate the performance of Algorithm 1 in a synthetic environment. Due to space constraints, we focus on the *stochastic* setting, that is, both the rewards and the constraints are sampled from fixed distributions, while we refer to Appendix D for the complete experimental evaluation. This choice is primarily motivated by the fact that the stochastic setting is indeed the hardest for algorithms capable of handling stochastic and adversarial constraints simultaneously. Indeed, stochastic environments allow us to employ strong algorithmic benchmarks, that is, algorithms tailored for stochastic settings only, to compare our algorithm with. Specifically, we consider the following algorithms. ($i$) OptCMDP (Algorithm 1 of (Efroni et al., 2020)). This algorithm solves an optimistic linear programming formulation of the CMDP, at each episode. OptCMDP attains $\widetilde{\mathcal{O}}(\sqrt{T})$ regret and *positive* violation, without Slater's condition, being arguably state-of-the-art in terms of performance for the stochastic setting. ($ii$) OptPrimalDual-CMDP (Algorithm 4 of (Efroni et al., 2020)). This algorithm employs a primal-dual approach, performing incremental updates for both the primal (that is, the policy) and dual Lagrange variables. OptPrimalDual-CMDP attains $\widetilde{\mathcal{O}}(\frac{1}{\rho}\sqrt{T})$ regret and violation, assuming Slater's condition. In Figure 1, we provide the results of our synthetic evaluation. Specifically, in Figure 1a, we provide the regret attained by Algorithm 1 and the aforementioned benchmarks. As expected, the performance of WC-OPS is comparable with the one of OptCMDP. Differently, OptPrimalDual-CMDP, which relies on the Slater's parameter of the problem, attains worse regret guarantees. Similarly, in Figure 1b, we provide the results in terms of constraints violation. In such a case, Algorithm 1 attains significantly better performance than both OptCMDP and OptPrimalDual-CMDP.

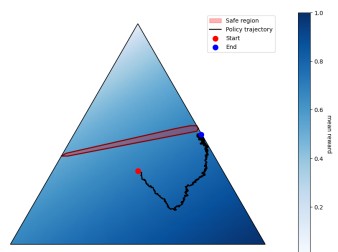

Figure 2: Trajectory of policy $\pi_t$

In Figure 2, we show the trajectory of the policy $\pi_t$ over a three-dimensional simplex in the case of a CMDP with a single state and three actions. The figure illustrates how Algorithm 1 asymptotically converges to the safe decision space, highlighted in red, while playing as much as possible the optimal action, which is shaded in blue.

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

CONTENTS

## A  ADDITIONAL RELATED WORKS

In this section, we provide a brief overview of the main research directions that are relevant to our work. We start by describing works dealing with the more general setting of MDPs and we proceed introducing constraints, first in the single-state case and then in the CMDP case.

### A.1  ONLINE LEARNING IN MDPs

MDPs have been widely employed as a framework to model decision-making problems, in particular in online settings. In such a context, different assumptions have been made about the type of feedback received by the learner and how the feedback is generated. Many works, such as Auer et al. (2008); Zimin & Neu (2013); Azar et al. (2017), consider bandit feedback, i.e. the algorithm only observes the loss/reward for the specific state-action pair visited. In contrast, works such as Even-Dar et al. (2009) and Rosenberg & Mansour (2019b) consider a full-information feedback, i.e. the algorithm receives the complete loss/reward information. Most of the first works on MDPs are set in a stochastic environment, i.e. the loss is assumed to be generated according to a certain (unknown) distribution (see Auer et al. (2008); Azar et al. (2017)). Other works, such as Even-Dar et al. (2009); Neu et al. (2010); Rosenberg & Mansour (2019a;b); Jin et al. (2020; 2024), consider feedback adversarially generated.

### A.2  ONLINE LEARNING WITH CONSTRAINTS

Various studies have been made about the single state bandit with constraints problem (Liakopoulos et al., 2019; Pacchiano et al., 2021). In such a case, best-of-both-worlds primal-dual algorithms, covering both stochastic and adversarial settings, were designed in Castiglioni et al. (2022a), Castiglioni et al. (2022b). Primal-dual methods have long been the only effective approach to tackle online learning problems in bandits with constraints, although they require strong assumptions. The first best-of-both-worlds solution for constrained bandits that does not rely on a primal-dual approach was proposed by Bernasconi et al. (2024).

### A.3  ONLINE LEARNING IN CMDPs

Online Learning in CMDPs has gained increasing attention recently, given its relevance in real-world applications, such as autonomous vehicles (Isele et al., 2018; Wen et al., 2020), bidding (Gummadi et al., 2012; Wu et al., 2018; He et al., 2021) and recommendation systems (Singh et al., 2020). The existing works about CMDPs cover both the case where the loss/reward is stochastic and the case where it is adversarially chosen. Specifically, Efroni et al. (2020) deals with finite-horizon CMDPs, with stochastic losses and constraints, unknown transition function and bandit feedback. The authors analyze two approaches, both providing sublinear regret and cumulative constraint violation. Stradi et al. (2025b) propose the first primal-dual algorithm capable of attaining sublinear positive violation in the stochastic setting, improving the results previously established in (Ghosh et al., 2024; Müller et al., 2024). Zheng & Ratliff (2020) studies episodic CMDPs with stochastic losses and constraints, known transition function and bandit feedback. This algorithm achieves $\widetilde{\mathcal{O}}(T^{\frac{3}{4}})$ regret and guarantees that the cumulative constraint violation remains below a certain threshold with a given probability. (Qiu et al., 2020) achieves sublinear regret and violation in episodic CMDPs with adversarial losses, stochastic constraints, unknown transition function and full-information feedback. Stradi et al. (2025a) proposes the first algorithm to handle CMDPs with adversarial losses and bandit feedback. The constraint functions considered in most of the works are stochastic. As for adversarial settings, Mannor et al. (2009) prove (for the easier single state setting) the impossibility of attaining both sublinear regret and constraint violation with respect to a policy that satisfies the constraints on average. The first best-of-both-worlds algorithm for online learning in episodic CMDPs was proposed by Stradi et al. (2024a), which employs a primal-dual approach providing $\widetilde{\mathcal{O}}(\sqrt{T})$ cumulative regret and constraint violation under a Slater-like satisfiability condition and $\widetilde{\mathcal{O}}(T^{\frac{3}{4}})$ regret and constraint violation without such a condition. This algorithm only works under

full-information feedback. Stradi et al. (2025c) overcomes this limitation by proving similar guarantees in a setting with bandit feedback, employing a primal-dual policy optimization method. Finally, Wei et al. (2023), Ding & Lavaei (2023) and Stradi et al. (2024b) consider the case in which rewards and constraints are non-stationary, assuming that their variation is bounded. Thus, their results are *not* applicable to general adversarial settings.

# B    OMITTED PROOFS AND LEMMAS OF SECTION 4

In this section, we provide the omitted proofs and the additional lemmas for the theoretical analysis of Algorithm 1.

## B.1    RESULTS ON THE OPTIMIZATION UPDATE

In this section, we provide the results associated to the optimization update performed by Algorithm 1. We start with the following lemma.

**Lemma B.1.** *For any $\delta \in (0,1)$ and for any $q \in \bigcap_{t \in [T]} \widehat{\Delta}_t(\mathcal{P}_t)$, Algorithm 1 attains:*

$$\sum_{t=1}^{T} \widehat{\ell}_t^\top (\widehat{q}_t - q) \le L \frac{\ln(|X|^2|A|)}{\eta} + \eta |X||A|T + \frac{\eta L \ln \frac{L}{\delta}}{\gamma},$$

*with probability at least $1 - \delta$.*

*Proof.* Define $\tilde{q}_{t+1}$ such that:

$$\tilde{q}_{t+1}(x, a, x') = \widehat{q}_t(x, a, x')e^{-\eta \widehat{\ell}_t(x,a)}.$$

Since it holds:

$$\widehat{q}_{t+1} = \underset{q \in \Delta(\mathcal{P}_t)}{\arg \min} B(q \| \tilde{q}_{t+1}),$$

and

$$\eta \widehat{\ell}_t^\top (\widehat{q}_t - q) = B(q \| \widehat{q}_t) - B(q \| \tilde{q}_{t+1}) + B(\widehat{q}_t \| \tilde{q}_{t+1}),$$

by the condition $q \in \bigcap_{t \in [T]} \widehat{\Delta}_t(\mathcal{P}_t)$ and the generalized Pythagorean theorem, it holds:

$$B(q \| \widehat{q}_t) \le B(q \| \tilde{q}_{t+1}).$$

Therefore, we have:

$$\eta \sum_{t=1}^{T} \widehat{\ell}_t^\top (\widehat{q}_t - q) = \sum_{t=1}^{T} B(q \| \widehat{q}_t) - B(q \| \widehat{q}_{t+1}) + B(\widehat{q}_t \| \tilde{q}_{t+1})$$

$$= B(q \| \widehat{q}_1) - B(q \| \widehat{q}_{T+1}) + \sum_{t=1}^{T} B(\widehat{q}_t \| \tilde{q}_{t+1})$$

$$\le B(q \| \widehat{q}_1) + \sum_{t=1}^{T} B(\widehat{q}_t \| \tilde{q}_{t+1}).$$

To bound the term $B(\widehat{q}_t \| \tilde{q}_{t+1})$, we proceed as follows:

$$B(\widehat{q}_t \| \tilde{q}_{t+1}) = \sum_{k=0}^{L-1} \sum_{x \in X_k, a \in A, x' \in X_{k+1}} \left( \eta \widehat{q}_t(x, a, x') \widehat{\ell}_t(x, a) + \widehat{q}_t(x, a, x') e^{-\eta \widehat{\ell}_t(x,a)} \right)$$

$$= \eta^2 \sum_{k=0}^{L-1} \sum_{x \in X_k, a \in A, x' \in X_{k+1}} \widehat{q}_t(x, a, x') \widehat{\ell}_t(x, a)^2$$

$$= \eta^2 \sum_{x \in X, a \in A} \widehat{q}_t(x, a) \widehat{\ell}_t(x, a)^2 \tag{5}$$

$$= \eta^2 \sum_{x \in X, a \in A} \frac{\ell_t(x,a)}{u_t(x,a) + \gamma} \widehat{\ell}_t(x,a)$$

$$\leq \eta^2 \sum_{x \in X, a \in A} \widehat{\ell}_t(x,a)$$

$$\leq \eta \sum_{x \in X, a \in A} \frac{q_t(x,a)}{u_t(x,a)} \ell_t(x,a) + \eta \frac{L \ln \frac{L}{\delta}}{2\gamma} \tag{6}$$

$$\leq \eta \sum_{x \in X, a \in A} \ell_t(x,a) + \eta \frac{L \ln \frac{L}{\delta}}{2\gamma}$$

$$\leq \eta |X||A|T + \eta \frac{L \ln \frac{L}{\delta}}{2\gamma}, \tag{7}$$

where Equation (5) is due to the fact that $1 - e^{-z} \leq z$ for all $z \geq 0$, Inequality (6) holds with probability at least $1 - \delta$ by Lemma 11 from (Jin et al., 2020) with $\alpha_t(x,a) = 2\gamma$.

Choosing $q_1(x,a,x') = \frac{1}{|X_k||A||X_{k+1}|}$ we have $B(q||q_1) \leq L \ln(|X|^2|A|)$ and therefore:

$$\eta \sum_{t=1}^T \widehat{\ell}_t^\top (\widehat{q}_t - q) \leq \ln(|X|^2|A|) + \eta^2 \sum_{x \in X, a \in A} q(x,a) \widehat{\ell}_t(x,a)^2$$

$$= L \ln(|X|^2|A|) + \eta^2 |X||A|T + \eta^2 \frac{L \ln \frac{L}{\delta}}{2\gamma}.$$

Rearranging, we obtain the following final result:

$$\sum_{t=1}^T \widehat{\ell}_t^\top (\widehat{q}_t - q) \leq L \frac{\ln(|X|^2|A|)}{\eta} + \eta |X||A|T + \eta \frac{L \ln \frac{L}{\delta}}{2\gamma}.$$

This concludes the proof. $\qquad\square$

We conclude by showing the following performance bound.

**Theorem B.2.** *For any* $\delta \in (0,1)$ *and for any* $q \in \bigcap_{t \in [T]} \widehat{\Delta}_t(\mathcal{P}_t)$, *Algorithm 1, with* $\eta = \gamma = \sqrt{\frac{L \ln\left(\frac{L|X||A|}{\delta}\right)}{T|X||A|}}$, *attains:*

$$\sum_{t=1}^T r_t^\top (q - q_t) \leq 14 L |X|^2 \sqrt{2T|A| \ln\left(\frac{T|X|^2|A|}{\delta}\right)},$$

*with probability at least* $1 - 15\delta$.

*Proof.* It holds:

$$\sum_{t=1}^T \ell_t^\top (q_t - q) = \sum_{t=1}^T (\ell_t - \widehat{\ell}_t)^\top \widehat{q}_t + \sum_{t=1}^T (\widehat{\ell}_t - \ell_t)^\top q + \sum_{t=1}^T \widehat{\ell}_t^\top (\widehat{q}_t - q) + \sum_{t=1}^T \ell_t^\top (q_t - \widehat{q}_t)$$

$$\leq \gamma |X||A|T + 2L|X|^2 \sqrt{2T \ln\left(\frac{L|X|}{\delta}\right)}$$

$$+ 3L|X|^2 \sqrt{2T|A| \ln\left(\frac{T|X|^2|A|}{\delta}\right)} + \sum_{t=1}^T (\widehat{\ell}_t - \ell_t)^\top q$$

$$+ \sum_{t=1}^T \widehat{\ell}_t^\top (\widehat{q}_t - q) + \sum_{t=1}^T \ell_t^\top (q_t - \widehat{q}_t) \tag{8}$$

$$\leq \gamma|X||A|T + 2L|X|^2\sqrt{2T\ln\left(\frac{L|X|}{\delta}\right)}$$

$$+ 3L|X|^2\sqrt{2T|A|\ln\left(\frac{T|X|^2|A|}{\delta}\right)} + \frac{L\ln(|X||A|/\delta)}{\gamma}$$

$$+ \sum_{t=1}^{T}\widehat{\ell}_t^{\top}(\widehat{q}_t - q) + \sum_{t=1}^{T}\ell_t^{\top}(q_t - \widehat{q}_t) \tag{9}$$

$$\leq \gamma|X||A|T + 2L|X|^2\sqrt{2T\ln\left(\frac{L|X|}{\delta}\right)}$$

$$+ 3L|X|^2\sqrt{2T|A|\ln\left(\frac{T|X|^2|A|}{\delta}\right)} + \frac{L\ln(|X||A|/\delta)}{\gamma}$$

$$+ L\frac{\ln(|X|^2|A|)}{\eta} + \eta|X||A|T + \frac{\eta L\ln\frac{L}{\delta}}{\gamma} + \sum_{t=1}^{T}\ell_t^{\top}(q_t - \widehat{q}_t) \tag{10}$$

$$\leq \gamma|X||A|T + 2L|X|^2\sqrt{2T\ln\left(\frac{L|X|}{\delta}\right)}$$

$$+ 3L|X|^2\sqrt{2T|A|\ln\left(\frac{T|X|^2|A|}{\delta}\right)} + \frac{L\ln(|X||A|/\delta)}{\gamma}$$

$$+ L\frac{\ln(|X|^2|A|)}{\eta} + \eta|X||A|T + \frac{\eta L\ln\frac{L}{\delta}}{\gamma}$$

$$+ 2L|X|\sqrt{2T\ln\frac{2L}{\delta}} + 3L|X|\sqrt{2T|A|\ln\frac{2T|X||A|}{\delta}} \tag{11}$$

$$\leq \sqrt{|X||A|TL\ln\left(\frac{L|X||A|}{\delta}\right)} + 2L|X|^2\sqrt{2T\ln\left(\frac{L|X|}{\delta}\right)}$$

$$+ 3L|X|^2\sqrt{2T|A|\ln\left(\frac{T|X|^2|A|}{\delta}\right)} + 3\sqrt{|X||A|TL\ln\left(\frac{L|X||A|}{\delta}\right)}$$

$$+ 2L|X|\sqrt{2T\ln\frac{2L}{\delta}} + 3L|X|\sqrt{2T|A|\ln\frac{2T|X||A|}{\delta}}$$

$$\leq (4 + 2 + 3 + 2 + 3)L|X|^2\sqrt{2T|A|\ln\left(\frac{T|X|^2|A|}{\delta}\right)}$$

$$= 14L|X|^2\sqrt{2T|A|\ln\left(\frac{T|X|^2|A|}{\delta}\right)},$$

where Inequality (8) holds by Lemma C.5 with probability $1 - 7\delta$, Inequality (9) holds by Lemma 14 of (Jin et al., 2020) with probability $1 - 5\delta$, Inequality (10) holds by Lemma B.1 with probability $1 - \delta$ and Inequality (11) holds by Lemma B.3 of (Rosenberg & Mansour, 2019b) with probability $1 - 2\delta$. By Union Bound, the final result holds with probability $1 - 15\delta$. Since by definition $\ell_t(x, a) = 1 - r_t(x, a)\mathbb{I}_t\{x, a\}$ for all $x \in X, a \in A$, it holds:

$$\sum_{t=1}^{T}\ell_t^{\top}(q_t - q) = \sum_{t=1}^{T}r_t^{\top}(q - q_t) \leq 14L|X|^2\sqrt{2T|A|\ln\left(\frac{T|X|^2|A|}{\delta}\right)},$$

which concludes the proof. $\square$

## B.2 Results on the Decision Space

In this section, we provide the results on the decision space definition of Algorithm 1.

We start by showing that, in the stochastic setting, the confidence bound plays a central role in the definition of the decision space.

**Theorem B.3.** *In the stochastic setting, let $\delta \in (0, 1)$ and $b_t(x, a)$ such that with probability at least $1 - \delta$, it holds $|\widehat{g}_{t,i}(x, a) - \bar{g}_i(x, a)| \leq b_t(x, a)$, for all $(x, a) \in X \times A, i \in [m], t \in [T]$. Furthermore, let $\Delta^\star = \left\{ q \in \Delta(M) : \bar{g}_i^\top q \leq 0, \forall i \in [m] \right\}$. Then, with probability at least $1 - 2\delta$ it holds:*

$$\Delta^\star \subseteq \widehat{\Delta}_t(\mathcal{P}_t), \; \forall t \in [T].$$

*Proof.* Assume the condition of the theorem holds. Let $q \in \Delta^\star$ and consider the following inequalities:

$$\begin{aligned}
\widehat{g}_{t,i}^\top q &= (\widehat{g}_{t,i} - \bar{g}_i)^\top q + \bar{g}_i^\top q \\
&\leq (\widehat{g}_{t,i} - \bar{g}_i)^\top q \\
&= \sum_{x \in X, a \in A} (\widehat{g}_{t,i}(x, a) - \bar{g}_i(x, a))q(x, a) \\
&\leq b_t^\top q,
\end{aligned}$$

where the first inequality holds by definition of $\Delta^\star$ and the second inequality follows from the definition of $b_t$. Thus, $(\widehat{g}_{t,i} - b_t)^\top q \leq 0$, which by definition proves that $q \in \widehat{\Delta}_t(\mathcal{P}_t)$. The final results follows from noticing that by Lemma 4.1 of (Rosenberg & Mansour, 2019b) $P \in \mathcal{P}_t$ with probability at least $1 - \delta$. A final union bound concludes the proof. □

We proceed by proving a similar result for the adversarial setting. The key insight here is that confidence bounds are not necessary.

**Theorem 4.5.** *In the adversarial setting, let $\delta \in (0, 1)$ and $\Delta^\diamond$ be the interpolation of any point $q \in \Delta(M)$ and $q^\diamond$ and let $\rho' = L \cdot \rho$. Formally,*

$$\Delta^\diamond := \frac{L}{L + \rho'}\{q^\diamond\} + \frac{\rho'}{L + \rho'}\Delta(M).$$

*Then, with probability at least $1 - \delta$, it holds that $\Delta^\diamond \subseteq \widehat{\Delta}_t(\mathcal{P}_t)$ for all $t \in [T]$.*

*Proof.* For each $t \in [T]$, $i \in [m]$, and $(x, a) \in X \times A$, it holds:

$$\widehat{g}_{t,i}(x, a) = \sum_{\tau \in \mathcal{T}_{t,x,a}} w_{t,x,a}(\tau)g_{\tau,i}(x, a),$$

and by the weights definition,

$$\sum_{\tau \in \mathcal{T}_{t,x,a}} w_{t,x,a}(\tau) = 1.$$

Thus notice that, for all $t \in [T]$ and constraint $i \in [m]$, we have:

$$\max_{(x,a) \in \mathcal{Q}(q^\diamond)} \widehat{g}_{t,i}(x, a)q^\diamond(x, a) \leq -\rho,$$

which implies:

$$\widehat{g}_{t,i}^\top q^\diamond \leq -L \cdot \rho = -\rho'.$$

Moreover notice that:

$$\widehat{g}_{t,i}^\top q \leq L, \quad \forall q \in \Delta(M).$$

Thus, for any $\tilde{q} \in \Delta^\diamond$ and $q \in \Delta(M)$, we obtain:

$$\begin{aligned}
\widehat{g}_{t,i}^\top \tilde{q} &= \frac{L}{L + \rho'} \widehat{g}_{t,i}^\top q^\diamond + \frac{\rho'}{L + \rho'} \widehat{g}_{t,i}^\top q \\
&\leq \frac{L}{L + \rho'}(-\rho') + \frac{\rho'}{L + \rho'}L \\
&\leq 0,
\end{aligned}$$

that is, $\tilde{q} \in \widehat{\Delta}_t(P)$. As in the stochastic case, the final result follows from noticing that $\Delta(M) \subseteq \Delta(\mathcal{P}_t)$ since, with probability at least $1 - \delta$, $P \in \mathcal{P}_t$, by Lemma 4.1 of (Rosenberg & Mansour, 2019b). □

## B.3 RESULTS ON THE WEIGHTS

In this section, we provide some fundamental results on the weights employed by Algorithm 1.

**Proposition 3.1.** *If $\beta_{t,i}(x,a) = \frac{1}{N_t(x,a)}$ for every $\tau \in \mathcal{T}_{t-1,x,a}$ such that $x_\tau = x, a_\tau = a$, then the following holds:*

$$w_{t,x,a,i}(\tau) = \frac{1}{N_t(x,a)},$$

*and we recover the empirical mean estimator:*

$$\widehat{g}_{t,i}(x,a) = \frac{1}{N_t(x,a)} \sum_{\tau \in \mathcal{T}_{t-1,x,a}} g_{\tau,i}(x,a).$$

*Proof.* Consider a pair $(x,a) \in X \times A$, an index $i \in [m]$, and $t \in [T]$. By applying Lemma 5.3 of (Bernasconi et al., 2024), we obtain:

$$w_{t,x,a,i}(\tau) = \beta_{\tau,i}(x,a) \prod_{h=\tau+1}^{t-1} (1 - \beta_{h,i}(x,a))$$

$$= \frac{1}{N_\tau(x,a)} \prod_{h \in \mathcal{T}_{t-1,x,a}:h>\tau} \left(1 - \frac{1}{N_h(x,a)}\right), \quad \forall \tau \in \mathcal{T}_{t-1,x,a}.$$

We now focus on the term $\prod_{h \in \mathcal{T}_{t-1,x,a}:h>\tau} \left(1 - \frac{1}{N_h(x,a)}\right)$:

$$\prod_{h \in \mathcal{T}_{t-1,x,a}:h>\tau} \left(1 - \frac{1}{N_h(x,a)}\right) = \prod_{h \in \mathcal{T}_{t-1,x,a}:h>\tau} \frac{N_h(x,a) - 1}{N_h(x,a)}$$

$$= \prod_{j=N_\tau(x,a)+1}^{N_{t-1}(x,a)} \frac{j-1}{j}$$

$$= \frac{N_\tau(x,a)}{N_{t-1}(x,a)}.$$

Thus:

$$w_{t,x,a,i}(\tau) = \frac{1}{N_\tau(x,a)} \frac{N_\tau(x,a)}{N_{t-1}(x,a)} = \frac{1}{N_{t-1}(x,a)}.$$

This concludes the proof. $\square$

We proceed by relating the violation attained by Algorithm 1 to the weighted estimators.

**Theorem B.4.** *Given an interval $[t_1, t_2] \subseteq [T]$, $i \in [m]$ and $\delta \in (0,1)$, Algorithm 1 attains the following bound with probability at least $1 - 3\delta$:*

$$V_{[t_1,t_2],i} \leq \sum_{x \in X, a \in A} \sum_{\tau \in \mathcal{T}_{t_2,x,a} \cap [t_1,t_2]} \frac{1}{\beta_{\tau,i}(x,a)} (\widehat{g}_{\tau+1,i}(x,a) - \widehat{g}_{\tau,i}(x,a)) + \sum_{\tau=t_1}^{t_2} b_\tau^\top \widehat{q}_\tau$$

$$+ 7L|X|\sqrt{2(t_2 - t_1)|A| \ln \frac{2T|X||A|}{\delta}},$$

*where $V_{[t_1,t_2],i} := \sum_{\tau=t_1}^{t_2} g_{\tau,i}^\top q_\tau$.*

*Proof.* It holds:

$$V_{[t_1,t_2],i} = \sum_{\tau=t_1}^{t_2} g_{\tau,i}^\top q_\tau$$

$$\leq \sum_{\tau=t_1}^{t_2} g_{\tau,i}^\top q_\tau + \sum_{\tau=t_1}^{t_2} b_\tau^\top \widehat{q}_\tau - \sum_{\tau=t_1}^{t_2} \widehat{g}_{\tau,i}^\top \widehat{q}_\tau \tag{12}$$

$$= \sum_{\tau=t_1}^{t_2} g_{\tau,i}^\top q_\tau + \sum_{\tau=t_1}^{t_2} b_\tau^\top \widehat{q}_\tau - \sum_{\tau=t_1}^{t_2} \widehat{g}_{\tau,i}^\top \widehat{q}_\tau + \sum_{\tau=t_1}^{t_2} \widehat{g}_{\tau,i}^\top q_\tau - \sum_{\tau=t_1}^{t_2} \widehat{g}_{\tau,i}^\top q_\tau$$

$$= \sum_{\tau=t_1}^{t_2} (g_{\tau,i} - \widehat{g}_{\tau,i})^\top q_\tau + \sum_{\tau=t_1}^{t_2} b_\tau^\top \widehat{q}_\tau + \sum_{\tau=t_1}^{t_2} \widehat{g}_{\tau,i}^\top (q_\tau - \widehat{q}_\tau)$$

$$\leq \sum_{\tau=t_1}^{t_2} \sum_{x \in X, a \in A} (g_{\tau,i}(x,a) - \widehat{g}_{\tau,i}(x,a)) \mathbb{I}_\tau \{x,a\} + 2L\sqrt{2(t_2 - t_1) \ln \frac{1}{\delta}}$$

$$+ \sum_{\tau=t_1}^{t_2} b_\tau^\top \widehat{q}_\tau + \|q_\tau - \widehat{q}_\tau\|_1 \tag{13}$$

$$\leq \sum_{\tau=t_1}^{t_2} \sum_{x \in X, a \in A} (g_{\tau,i}(x,a) - \widehat{g}_{\tau,i}(x,a)) \mathbb{I}_\tau \{x,a\} + 2L\sqrt{2(t_2 - t_1) \ln \frac{1}{\delta}} + \sum_{\tau=t_1}^{t_2} b_\tau^\top \widehat{q}_\tau$$

$$+ 2L|X|\sqrt{2(t_2 - t_1) \ln \frac{2L}{\delta}} + 3L|X|\sqrt{2(t_2 - t_1)|A| \ln \frac{2T|X||A|}{\delta}} \tag{14}$$

$$= \sum_{x \in X, a \in A} \sum_{\tau \in \mathcal{T}_{t_2,x,a} \cap [t_1,t_2]} \frac{(\widehat{g}_{\tau+1,i}(x,a) - \widehat{g}_{\tau,i}(x,a))}{\beta_{\tau,i}(x,a)}$$

$$+ \sum_{\tau=t_1}^{t_2} b_\tau^\top \widehat{q}_\tau + 2L|X|\sqrt{2(t_2 - t_1) \ln \frac{2L}{\delta}}$$

$$+ 3L|X|\sqrt{2(t_2 - t_1)|A| \ln \frac{2T|X||A|}{\delta}} + 2L\sqrt{2(t_2 - t_1) \ln \frac{1}{\delta}} \tag{15}$$

$$\leq \sum_{x \in X, a \in A} \sum_{\tau \in \mathcal{T}_{t_2,x,a} \cap [t_1,t_2]} \frac{(\widehat{g}_{\tau+1,i}(x,a) - \widehat{g}_{\tau,i}(x,a))}{\beta_{\tau,i}(x,a)}$$

$$+ \sum_{\tau=t_1}^{t_2} b_\tau^\top \widehat{q}_\tau + 7L|X|\sqrt{2(t_2 - t_1)|A| \ln \frac{2T|X||A|}{\delta}},$$

where Equation (12) is due to the fact that $\widehat{q}_\tau \in \widehat{\Delta}_\tau(\mathcal{P}_\tau)$, Inequality (13) holds by Lemma C.3, from which we have, with probability $1 - \delta$:

$$\sum_{\tau=t_1}^{t_2} (g_{\tau,i} - \widehat{g}_{\tau,i})^\top q_\tau \leq \sum_{\tau=t_1}^{t_2} \sum_{x \in X, a \in A} (g_{\tau,i}(x,a) - \widehat{g}_{\tau,i}(x,a)) \mathbb{I}_\tau \{x,a\} + 2L\sqrt{2(t_2 - t_1) \ln \frac{1}{\delta}}.$$

Inequality (14) follows from Lemma B.3 of (Rosenberg & Mansour, 2019b), with probability at least $1 - 2\delta$—notice that, all the results mentioned above can be trivially extended to hold in the interval $[t_1, t_2]$—. Equation (15) holds by the definition of the update:

$$\widehat{g}_{\tau,i}(x,a) = (1 - \beta_{\tau,i}(x,a)) \widehat{g}_{\tau,i}(x,a) + \beta_{\tau,i}(x,a) g_{\tau,i}(x,a), \quad \forall (x,a) \text{ such that } \mathbb{I}_\tau\{x,a\} = 1.$$

A final Union Bound concludes the proof. $\square$

We proceed with the following corollary.

**Corollary B.5.** *Given an interval $[t_1, t_2] \subseteq [T]$, $i \in [m]$ and $\delta > 0$, assume that for any $(x,a) \in X \times A$ it holds $\beta_{\tau,i}(x,a) \geq \beta_{\tau',i}(x,a)$ for each $\tau' \leq \tau \in \mathcal{T}_{t_2,x,a} \cap [t_1, t_2]$. Then, with probability at least $1 - 3\delta$ it holds:*

$$V_{[t_1,t_2],i} \leq \sum_{x \in X, a \in A} \frac{2}{\beta_{\ell(x,a,[t_1,t_2]),i}(x,a)} + \sum_{\tau=t_1}^{t_2} b_\tau^\top \widehat{q}_\tau + 7L|X|\sqrt{2(t_2 - t_1)|A| \ln \frac{2T|X||A|}{\delta}},$$

*where $\ell(x, a, [t_1, t_2])$ are the last rounds in the interval $[t_1, t_2]$ in which the pair $(x,a)$ is visited.*

*Proof.* Assuming Theorem B.4 holds with probability $1 - 3\delta$, it is sufficient to show:

$$\sum_{x \in X, a \in A} \sum_{\tau \in \mathcal{T}_{t_2,x,a} \cap [t_1, t_2]} \frac{1}{\beta_{\tau,i}(x,a)} \left(\widehat{g}_{\tau+1,i}(x,a) - \widehat{g}_{\tau,i}(x,a)\right) \leq \sum_{x \in X, a \in A} \sum_{\tau \in \mathcal{T}_{t_2,x,a} \cap [t_1, t_2]} \frac{1}{\beta_{t_2,i}(x,a)}$$

Fixing a $(x,a) \in X \times A$ and defining $h = |\mathcal{T}_{t_2,x,a} \cap [t_1, t_2]|$ as the number of times the pair $(x,a)$ is visited in the interval $[t_1, t_2]$, let $\tau(j)$ be the rounds in which the pair $(x,a)$ is visited the $j^{\text{th}}$ time in $[t_1, t_2]$. Then we have:

$$\sum_{\tau \in \mathcal{T}_{t_2,x,a} \cap [t_1, t_2]} \frac{1}{\beta_{\tau,i}(x,a)} \left(\widehat{g}_{\tau+1,i}(x,a) - \widehat{g}_{\tau,i}(x,a)\right)$$

$$= \sum_{j \in [h]} \frac{1}{\beta_{\tau(j),i}(x,a)} \left(\widehat{g}_{\tau(j+1),i}(x,a) - \widehat{g}_{\tau(j),i}(x,a)\right)$$

$$= \sum_{j \in [h-1]} \left(\frac{1}{\beta_{\tau(j),i}(x,a)} \left(\widehat{g}_{\tau(j+1),i}(x,a) - \widehat{g}_{\tau(j),i}(x,a)\right)\right)$$

$$+ \frac{1}{\beta_{\tau(h),i}(x,a)} \left(\widehat{g}_{\tau(h)+1,i}(x,a) - \widehat{g}_{\tau(h),i}(x,a)\right)$$

$$\leq \sum_{j \in [h-1]} \left(\frac{1}{\beta_{\tau(j+1),i}(x,a)} \widehat{g}_{\tau(j+1),i}(x,a) - \frac{1}{\beta_{\tau(j),i}(x,a)} \widehat{g}_{\tau(j),i}(x,a)\right)$$

$$+ \frac{1}{\beta_{\tau(h),i}(x,a)} \left(\widehat{g}_{\tau(h)+1,i}(x,a) - \widehat{g}_{\tau(h),i}(x,a)\right) \quad (16)$$

$$= \frac{1}{\beta_{\tau(h),i}(x,a)} \widehat{g}_{\tau(h)+1,i}(x,a) - \frac{1}{\beta_{\tau(1),i}(x,a)} \widehat{g}_{\tau(1),i}(x,a) \quad (17)$$

$$\leq \frac{2}{\beta_{\tau(h),i}(x,a)} \quad (18)$$

$$= \frac{2}{\beta_{\ell(x,a,[t_1,t_2]),i}(x,a)},$$

where Inequality (16) and Inequality (18) follow from the hypothesis that the learning rates are decreasing in the interval and Equation (17) follows from evaluating the telescoping sum. $\square$

### B.4 CONCENTRATION RESULTS

In this section, we provide a fundamental result on the concentration of the confidence bounds parameter employed by Algorithm 1. This is done in the following lemma.

**Lemma B.6.** *Given $c > 0$, $\alpha \in (0,1)$, $t \in [T]$ and $\delta \in (0,1)$, let $b_t(x,a) = \frac{c}{N_t(x,a)^\alpha}$ for all $(x,a) \in X \times A$. Then, with probability $1 - 3\delta$ it holds:*

$$\sum_{\tau=1}^{t} b_\tau^\top \widehat{q}_\tau \leq \frac{c}{1-\alpha} |X|^\alpha |A|^\alpha L^{1-\alpha} t^{1-\alpha} + 7L|X| \sqrt{2t|A| \ln \frac{2T|X||A|}{\delta}}$$

*Proof.* It holds:

$$\sum_{\tau=1}^{t} \sum_{x \in X, a \in A} b_\tau(x,a) \mathbb{I}_\tau\{x,a\} = c \sum_{\tau=1}^{t} \sum_{x \in X, a \in A} \frac{1}{N_\tau(x,a)^\alpha} \mathbb{I}_\tau\{x,a\}$$

$$= c \sum_{x \in X, a \in A} \sum_{h=1}^{N_t(x,a)} \frac{1}{h^\alpha}$$

$$\leq \frac{c}{1-\alpha} \sum_{x \in X, a \in A} N_t(x,a)^{1-\alpha}$$

$$\leq \frac{c}{1-\alpha}|X|^{\alpha}|A|^{\alpha}L^{1-\alpha}t^{1-\alpha}, \tag{19}$$

where Inequality (19) holds by Jensen's inequality. Moreover notice that:

$$\sum_{\tau=1}^{t} b_\tau^\top \widehat{q}_\tau = \sum_{\tau=1}^{t} b_\tau^\top \widehat{q}_\tau + \sum_{\tau=1}^{t} b_\tau^\top q_\tau - \sum_{\tau=1}^{t} b_\tau^\top q_\tau$$

$$\leq \|q_\tau - \widehat{q}_\tau\|_1 + \sum_{\tau=1}^{t} b_\tau^\top q_\tau$$

$$\leq 2L|X|\sqrt{2T\ln\frac{2L}{\delta}} + 3L|X|\sqrt{2T|A|\ln\frac{2T|X||A|}{\delta}} + \sum_{\tau=1}^{t} b_\tau^\top q_\tau \tag{20}$$

$$\leq 2L|X|\sqrt{2T\ln\frac{2L}{\delta}} + 3L|X|\sqrt{2T|A|\ln\frac{2T|X||A|}{\delta}}$$
$$+ 2L\sqrt{2T\ln\frac{1}{\delta}} + \sum_{\tau=1}^{t}\sum_{x\in X, a\in A} b_\tau(x,a)\mathbb{I}_\tau\{x,a\} \tag{21}$$

$$\leq 2L|X|\sqrt{2T\ln\frac{2L}{\delta}} + 3L|X|\sqrt{2T|A|\ln\frac{2T|X||A|}{\delta}}$$
$$+ 2L\sqrt{2T\ln\frac{1}{\delta}} + \frac{c}{1-\alpha}|X|^{\alpha}|A|^{\alpha}L^{1-\alpha}t^{1-\alpha} \tag{22}$$

$$\leq 7L|X|\sqrt{2T|A|\ln\frac{2T|X||A|}{\delta}} + \frac{c}{1-\alpha}|X|^{\alpha}|A|^{\alpha}L^{1-\alpha}t^{1-\alpha},$$

where Inequality (20) follows from Lemma B.3 of (Rosenberg & Mansour, 2019b), with probability at least $1-2\delta$, Inequality (21) holds by Lemma C.3 with probability at least $1-\delta$, Inequality (22) follows from Inequality (19). A Union Bound concludes the proof. $\qquad\square$

### B.5 VIOLATION BOUND

In this section, we provide the violation bound of Algorithm 1.

**Theorem B.7.** *Let $\delta \in (0,1)$. Both in stochastic and adversarial setting, with probability at least $1-4\delta$, Algorithm 1 attains:*

$$V_t \leq 61L|X|\sqrt{2t|A|\ln\left(\frac{2mT^2|X||A|}{\delta}\right)},$$

*for all $t \in [T]$.*

*Proof.* Given an $i \in [m]$, we assume that Corollary B.5 holds with probability $1-3\delta$ for any interval.

If $V_{t,i} \leq 61L\sqrt{|X||A|t\ln\left(\frac{T^2}{\delta}\right)}$ then the statement is trivially satisfied.

Otherwise, let us suppose that there exists a $\bar{t} \in T$ for which $V_{\bar{t},i} \geq 61L\sqrt{|X||A|t\ln\left(\frac{T^2}{\delta}\right)}$. This implies that there exists a $\underline{t} < \bar{t}$ such that $V_{t,i} \geq 44L\sqrt{|X||A|t\ln\left(\frac{T^2}{\delta}\right)}$ for all $t \in [\underline{t}, \bar{t}]$ and $V_{\underline{t}-1,i} \leq 44L\sqrt{|X||A|t\ln\left(\frac{T^2}{\delta}\right)}$. By Lemma C.3 it holds:

$$V_{t,i} = \sum_{\tau\in[t]} g_{\tau,i}^\top q_\tau \leq \sum_{\tau\in[t]}\sum_{x,a} g_{\tau,i}(x,a)\mathbb{I}_\tau\{x,a\} + 2L\sqrt{2t\ln\frac{1}{\delta}}.$$

with probability at least $1-\delta$. Therefore, since $V_{t,i} \geq 44L\sqrt{|X||A|t\ln\left(\frac{T^2}{\delta}\right)}$ for all $t \in [\underline{t}, \bar{t}]$, it holds:

$$\sum_{\tau\in[t]}\sum_{x,a} g_{\tau,i}(x,a)\mathbb{I}_\tau\{x,a\} \geq 44L\sqrt{|X||A|t\ln\left(\frac{T^2}{\delta}\right)} - 2L\sqrt{2t\ln\frac{1}{\delta}}$$

$$\geq 44L\sqrt{|X||A|t\ln\left(\frac{T^2}{\delta}\right)} - 2L\sqrt{|X||A|t\ln\left(\frac{T^2}{\delta}\right)}$$

$$= 42L\sqrt{|X||A|t\ln\left(\frac{T^2}{\delta}\right)}.$$

Thus, we can write:

$$\sum_{\tau\in[t]}\sum_{x,a}g_{\tau,i}(x,a)\mathbb{I}_\tau\{x,a\} - 21L|X|\sqrt{2t|A|\ln\frac{2mT^2|X||A|}{\delta}}$$

$$\geq 42L|X|\sqrt{2t|A|\ln\frac{2mT^2|X||A|}{\delta}} - 21L|X|\sqrt{2t|A|\ln\frac{2mT^2|X||A|}{\delta}}$$

$$\geq 21L|X|\sqrt{2t|A|\ln\frac{2mT^2|X||A|}{\delta}}.$$

and thus $\Gamma_{t,i} = 21L|X|\sqrt{2t|A|\ln\frac{2mT^2|X||A|}{\delta}}$ for all $t\in[\underline{t},\bar{t}]$.

Therefore on $t\in[\underline{t},\bar{t}]$ the learning rate can be lower-bounded as:

$$\beta_{t,i}(x,a) = \frac{(1+\Gamma_t)}{N_t(x,a)} = \frac{1+21L|X|\sqrt{2t|A|\ln\frac{2mT^2|X||A|}{\delta}}}{N_t(x,a)} \geq 21L|X|\sqrt{\frac{2|A|\ln\frac{2mT^2|X||A|}{\delta}}{N_t(x,a)}},$$

exploiting the fact that $N_t(x,a)\leq t$ for all $t\in[T]$. Therefore, by Corollary B.5 we can write:

$$V_{[\underline{t},\bar{t}],i} \leq \frac{2}{21L\sqrt{|X||A|t\ln\left(\frac{T^2}{\delta}\right)}}\sum_{x\in X,a\in A}\sqrt{N_{\bar{t}}(x,a)} + \sum_{\tau=\underline{t}}^{\bar{t}}b_\tau^\top\widehat{q}_\tau + 7L|X|\sqrt{2t|A|\ln\frac{2T^2|X||A|}{\delta}}$$

$$\leq \frac{2\sqrt{|X||A|L\bar{t}}}{21L\sqrt{|X||A|t\ln\left(\frac{T^2}{\delta}\right)}} + \sum_{\tau=\underline{t}}^{\bar{t}}b_\tau^\top\widehat{q}_\tau + 7L|X|\sqrt{2t|A|\ln\frac{2T^2|X||A|}{\delta}} \quad (23)$$

$$\leq \frac{2\sqrt{|X||A|L\bar{t}}}{21L\sqrt{|X||A|t\ln\left(\frac{T^2}{\delta}\right)}} + 2\sqrt{2|X||A|Lt\ln\left(\frac{2T^2|X||A|}{\delta}\right)}$$

$$\quad\quad + 14L|X|\sqrt{2t|A|\ln\frac{2T^2|X||A|}{\delta}} \quad (24)$$

$$\leq \left(\frac{1}{10} + 2 + 14\right)L|X|\sqrt{2t|A|\ln\left(\frac{2T^2|X||A|}{\delta}\right)},$$

where Inequality (23) holds by Jensen's Inequality and Inequality (24) holds by Lemma B.6, under the same event of Corollary B.5. Thus, we have:

$$V_{\bar{t},i} \leq V_{\underline{t},i} + V_{[\underline{t},\bar{t}],i}$$

$$\leq \left(44 + \frac{1}{10} + 2 + 14\right)L|X|\sqrt{2t|A|\ln\left(\frac{2T^2|X||A|}{\delta}\right)}$$

$$< 61L|X|\sqrt{2t|A|\ln\left(\frac{2T^2|X||A|}{\delta}\right)}.$$

This shows a contradiction, so there is no such $\bar{t}$. Taking a Union Bound on all $i\in[m]$ concludes the proof. $\qquad\square$

## B.6 Towards the Regret Bound in the Stochastic Setting

In this section we provide some preliminary results for the stochastic setting. Specifically, throughout the section we show that the violations are kept small during the learning dynamic, thus making

$\widehat{g}_{t,i}$ the empirical mean estimator of the constraints functions. This step is fundamental to show that the decision space of Algorithm 1 is suited to guarantee sublinear regret.

**Lemma B.8.** *Let $\delta \in (0, 1)$. With probability at least $1 - 2\delta$ it holds:*

$$V_{t,i} \leq \sum_{\tau=1}^{t} g_{\tau,i}^{\top} \widehat{q}_{\tau} + 5L|X|\sqrt{2T|A| \ln \frac{2mT|X||A|}{\delta}}, \quad \forall t \in [T], i \in [m].$$

*Proof.* It holds:

$$\begin{aligned}
V_{t,i} &= \sum_{\tau=1}^{t} g_{\tau,i}^{\top} q^{P,\pi_\tau} \\
&= \sum_{\tau=1}^{t} g_{\tau,i}^{\top} q^{P,\pi_\tau} + \sum_{\tau=1}^{t} g_{\tau,i}^{\top} \widehat{q}_\tau - \sum_{\tau=1}^{t-1} g_{\tau,i}^{\top} \widehat{q}_\tau \\
&\leq \sum_{\tau=1}^{t} g_{\tau,i}^{\top} \widehat{q}_\tau + \|q_\tau - \widehat{q}_\tau\|_1 \\
&\leq \sum_{\tau=1}^{t} g_{\tau,i}^{\top} \widehat{q}_\tau + 2L|X|\sqrt{2t \ln \frac{2L}{\delta}} + 3L|X|\sqrt{2t|A| \ln \frac{2T|X||A|}{\delta}} \quad (25) \\
&\leq \sum_{\tau=1}^{t} g_{\tau,i}^{\top} \widehat{q}_\tau + (2+3)L|X|\sqrt{2t|A| \ln \frac{2T|X||A|}{\delta}} \\
&= \sum_{\tau=1}^{t} g_{\tau,i}^{\top} \widehat{q}_\tau + 5L|X|\sqrt{2t|A| \ln \frac{2T|X||A|}{\delta}},
\end{aligned}$$

where Inequality (25) follows from Lemma B.3 of (Rosenberg & Mansour, 2019b), with probability at least $1 - 2\delta$. □

**Lemma B.9.** *Let $\delta \in (0, 1)$. With probability at least $1 - 3\delta$ it holds:*

$$\sum_{\tau=1}^{t} \bar{g}_i^{\top} \widehat{q}_\tau \leq \sum_{\tau=1}^{t} \sum_{x \in X, a \in A} \bar{g}_i(x,a) \mathbb{I}_\tau\{x,a\} + 7L|X|\sqrt{2t|A| \ln \frac{2mT|X||A|}{\delta}}, \quad \forall t \in [T], i \in [m].$$

*Proof.* It holds:

$$\begin{aligned}
\sum_{\tau=1}^{t} \bar{g}_i^{\top} \widehat{q}_\tau &= \sum_{\tau=1}^{t} \bar{g}_i^{\top} \widehat{q}_\tau + \sum_{\tau=1}^{t} \bar{g}_i^{\top} q^{P,\pi_\tau} - \sum_{\tau=1}^{t} \bar{g}_i^{\top} q^{P,\pi_\tau} \\
&\leq \sum_{\tau=1}^{t} \bar{g}_i^{\top} q^{P,\pi_\tau} + \|q_\tau - \widehat{q}_\tau\|_1 \\
&\leq \sum_{\tau=1}^{t} \bar{g}_i^{\top} q^{P,\pi_\tau} + 2L|X|\sqrt{2t \ln \frac{2L}{\delta}} + 3L|X|\sqrt{2t|A| \ln \frac{2T|X||A|}{\delta}} \quad (26) \\
&\leq \sum_{\tau=1}^{t} \sum_{x \in X, a \in A} \bar{g}_i(x,a) \mathbb{I}_\tau\{x,a\} + 2L\sqrt{2t \ln \frac{1}{\delta}} \\
&\quad + 2L|X|\sqrt{2t \ln \frac{2L}{\delta}} + 3L|X|\sqrt{2t|A| \ln \frac{2T|X||A|}{\delta}} \quad (27) \\
&\leq \sum_{\tau=1}^{t} \sum_{x \in X, a \in A} \bar{g}_i(x,a) \mathbb{I}_\tau\{x,a\} + (2+2+3)L|X|\sqrt{2t|A| \ln \frac{2T|X||A|}{\delta}} \\
&= \sum_{\tau=1}^{t} \sum_{x \in X, a \in A} \bar{g}_i(x,a) \mathbb{I}_\tau\{x,a\} + 7L|X|\sqrt{2t|A| \ln \frac{2T|X||A|}{\delta}},
\end{aligned}$$

where Inequality (26) follows from Lemma B.3 of (Rosenberg & Mansour, 2019b) with probability at least $1 - 2\delta$, Inequality (27) follows from Lemma C.3, with probability at least $1 - 2\delta$. A Union Bound concludes the proof. □

We conclude the section with the following lemma and the associated corollary, which allow us to state that the employment of the bonus quantity $b_t$ is necessary and sufficient to attain sublinear regret (and violation).

**Lemma 4.1.** *Let $\delta \in (0, 1)$. In the stochastic setting, with probability at least $1 - 11\delta$, it holds that:*

$$|\widehat{g}_{t,i}(x,a) - \bar{g}_i(x,a)| \le b_t(x,a) \quad \forall (x,a) \in X \times A, i \in [m], t \in [T].$$

*Proof.* To get the final result, it is sufficient to prove that for each $t \in [T]$ and $i \in [m]$, it holds:

$$\sum_{\tau \in [t]} \sum_{x,a} g_{\tau,i}(x,a) \mathbb{I}_\tau\{x,a\} \le 21L|X|\sqrt{2t|A|\ln\frac{2mT|X||A|}{\delta}}.$$

Our proof works by induction on $t$. It is trivial to show the inequality holds for $t = 1$. Indeed,

$$\sum_{x,a} g_{1,i}(x,a)\mathbb{I}_1\{x,a\} \le L \le 21L|X|\sqrt{2|A|\ln\frac{2mT|X||A|}{\delta}}.$$

Assuming that the inequality holds for all $\tau \le t - 1$, we now show that it holds also for $t$. By definition of $\Gamma_{\tau,i}$, the induction assumption implies that for $\tau \le t - 1$, we have $\beta_{\tau,i}(x,a) = \frac{1}{N_\tau(x,a)}$ for all $(x,a) \in X \times A, i \in [m]$. Then by Proposition 3.1 we have that:

$$\widehat{g}_{\tau,i}(x,a) = \frac{1}{N_\tau(x,a)} \sum_{\widehat{t} \in \mathcal{T}_{\tau,a}} g_{\widehat{t},i}(x,a)$$

Hence, by Lemma C.1, it holds, with probability at least $1 - \delta$:

$$|\widehat{g}_{\tau,i}(x,a) - \bar{g}_i(x,a)| \le \sqrt{\frac{2\ln\frac{2m|X||A|T}{\delta}}{N_t(x,a)}}, \quad \forall (x,a) \in X \times A, \tau \le t - 1.$$

Assuming that the event above holds, we consider the following inequalities:

$$\sum_{\tau \in [t]} \sum_{x,a} g_{\tau,i}(x,a)\mathbb{I}_\tau\{x,a\}$$

$$\le V_{t,i} + 2L\sqrt{2t\ln\frac{1}{\delta}} \tag{28}$$

$$= V_{t-1,i} + g_{t,i}^\top q_t + 2L\sqrt{2t\ln\frac{1}{\delta}}$$

$$\le \sum_{\tau=1}^{t-1} g_{\tau,i}^\top \widehat{q}_\tau + g_{t,i}^\top q_t + 2L\sqrt{2t\ln\frac{1}{\delta}} + 5L|X|\sqrt{2t|A|\ln\frac{2mT|X||A|}{\delta}} \tag{29}$$

$$\le \sum_{\tau=1}^{t-1} (g_{\tau,i} - \widehat{g}_{\tau,i})^\top \widehat{q}_\tau + \sum_{\tau=1}^{t-1} b_\tau^\top \widehat{q}_\tau + g_{t,i}^\top q_t + 2L\sqrt{2t\ln\frac{1}{\delta}} + 5L|X|\sqrt{2t|A|\ln\frac{2mT|X||A|}{\delta}} \tag{30}$$

$$\le \sum_{\tau=1}^{t-1} (g_{\tau,i} - \widehat{g}_{\tau,i})^\top \widehat{q}_\tau + 2\sqrt{2|X||A|Lt\ln\frac{2T|X||A|}{\delta}} + g_{t,i}^\top q_t$$

$$\qquad + 2L\sqrt{2t\ln\frac{1}{\delta}} + 12L|X|\sqrt{2t|A|\ln\frac{2mT|X||A|}{\delta}} \tag{31}$$

$$\le \sum_{\tau=1}^{t-1} (g_{\tau,i} - \widehat{g}_{\tau,i})^\top \widehat{q}_\tau + 2\sqrt{2|X||A|Lt\ln\frac{2T|X||A|}{\delta}} + L$$

$$\qquad + 2L\sqrt{2t\ln\frac{1}{\delta}} + 12L|X|\sqrt{2t|A|\ln\frac{2mT|X||A|}{\delta}}$$

$$\leq \sum_{\tau=1}^{t-1} (\bar{g}_i - \hat{g}_{\tau,i})^\top \hat{q}_\tau + 2\sqrt{2|X||A|Lt \ln \frac{2T|X||A|}{\delta}} + L$$

$$+ 12L|X|\sqrt{2t|A| \ln \frac{2mT|X||A|}{\delta}} + 4L\sqrt{2t \ln \frac{1}{\delta}} \tag{32}$$

$$\leq \sum_{\tau=1}^{t-1} \sum_{x \in X, a \in A} (\bar{g}_i(x,a) - \hat{g}_{\tau,i}(x,a)) \, \mathbb{I}_\tau\{x,a\} + 2\sqrt{2|X||A|Lt \ln \frac{2T|X||A|}{\delta}} + L$$

$$+ 12L|X|\sqrt{2t|A| \ln \frac{2mT|X||A|}{\delta}} + 4L\sqrt{2t \ln \frac{1}{\delta}} \tag{33}$$

$$\leq \sqrt{2 \ln \frac{2m|X||A|T}{\delta}} \sum_{x \in X, a \in A} \sum_{\tau=1}^{t-1} \frac{1}{\sqrt{N_\tau(x,a)}} \mathbb{I}_\tau\{x,a\} + 2\sqrt{2|X||A|Lt \ln \frac{2T|X||A|}{\delta}} + L$$

$$+ 12L|X|\sqrt{2t|A| \ln \frac{2mT|X||A|}{\delta}} + 4L\sqrt{2t \ln \frac{1}{\delta}}$$

$$\leq 2\sqrt{2|X||A|t \ln \frac{2m|X||A|T}{\delta}} + 2\sqrt{2|X||A|Lt \ln \frac{2T|X||A|}{\delta}} + L$$

$$+ 12L|X|\sqrt{2t|A| \ln \frac{2mT|X||A|}{\delta}} + 4L\sqrt{2t \ln \frac{1}{\delta}}$$

$$\leq (4 + 1 + 12 + 4)L|X|\sqrt{2t|A| \ln \frac{2mT|X||A|}{\delta}},$$

where Inequality (28) holds by Lemma C.3 with probability $1 - \delta$, Inequality (29) holds by Lemma B.8 with probability at least $1 - 2\delta$, Inequality (30) holds because $\hat{q}_\tau \in \hat{\Delta}_\tau(\mathcal{P}_\tau)$, Inequality (31) holds by Lemma B.6 with probability at least $1 - 3\delta$, taking $\alpha = \frac{1}{2}$ and $c = \sqrt{2 \ln \left( \frac{2|X||A|T}{\delta} \right)}$, Inequality (32) holds by Lemma C.4 with probability at least $1 - \delta$, Inequality (33) holds by Lemma B.9 with probability at least $1 - 3\delta$.

Thus $\sum_{\tau \in [t]} \sum_{x,a} g_{\tau,i}(x,a) \mathbb{I}_\tau\{x,a\} \leq 21L|X|\sqrt{2t|A| \ln \frac{2mT|X||A|}{\delta}}$, $\Gamma_{t,i} = 0$ and $\hat{g}_{t,i}(x,a)$ is the empirical mean of past observations. Therefore, by Lemma C.1 we have with probability at least $1 - \delta$:

$$|\hat{g}_{t,i}(x,a) - \bar{g}_i(x,a)| \leq \sqrt{\frac{2 \ln \left( \frac{2|X||A|mT}{\delta} \right)}{N_t(x,a)}} \quad \forall (x,a) \in X \times A, i \in [m], t \in [T].$$

A final Union Bound concludes the proof. $\qquad \square$

Thus, the following corollary holds.

**Corollary 4.2.** *In the stochastic setting, let $\delta \in (0,1)$ and $\Delta^\star = \left\{ q \in \Delta(M) : \bar{g}_i^\top q \leq 0 \; \forall i \in [m] \right\}$. Then, with probability at least $1 - 11\delta$, it holds:*

$$\Delta^\star \subseteq \hat{\Delta}_t(\mathcal{P}_t) \quad \forall t \in [T].$$

### B.7 Final Results

We provide the theoretical guarantees of Algorithm 1 in the stochastic setting.

**Theorem 4.3.** *Let $\delta \in (0,1)$. In the stochastic setting, Algorithm 1, with $\eta = \gamma = \sqrt{\frac{L \ln(L|X||A|/\delta)}{T|X||A|}}$, guarantees that with probability at least $1 - 30\delta$:*

$$R_T \leq 14L|X|^2 \sqrt{2T|A| \ln \left( \frac{T|X|^2|A|}{\delta} \right)},$$

*and*

$$V_t \leq 61L|X|\sqrt{2t|A|\ln\left(\frac{2mT^2|X||A|}{\delta}\right)}, \quad \forall t \in [T].$$

*Proof.* By Corollary 4.2 with probability at least $1 - 11\delta$, it holds $\Delta^\star \subseteq \cap_{t \in [T]} \widehat{\Delta}_t(\mathcal{P}_t)$. By Theorem B.2, we have that for any $q \in \bigcap_{t \in [T]} \widehat{\Delta}_t(\mathcal{P}_t)$, with probability at least $1 - 15\delta$, it holds:

$$\sum_{t \in [T]} r_t^\top (q - q_t) \leq 14L|X|^2\sqrt{2T|A|\ln\left(\frac{T|X|^2|A|}{\delta}\right)}.$$

Let $q^* = \arg\max_{q \in \Delta^\star} \sum_{t=1}^T r_t^\top q$. Then, by Union Bound we have that with probability at least $1 - 26\delta$ it holds:

$$\sum_{t \in [T]} r_t^\top (q^* - q_t) \leq 14L|X|^2\sqrt{2T|A|\ln\left(\frac{T|X|^2|A|}{\delta}\right)},$$

Similarly, with probability at least $1 - 4\delta$ by Theorem B.7:

$$V_t \leq 61L|X|\sqrt{2t|A|\ln\left(\frac{2mT^2|X||A|}{\delta}\right)}$$

By a Union Bound on all the events, this holds with probability at least $1 - 30\delta$.

This concludes the proof. $\qquad\square$

We conclude the section by providing the theoretical guarantees of Algorithm 1 in the adversarial setting.

**Theorem 4.6.** *Let $\delta \in (0,1)$. In the adversarial setting, Algorithm 1, with $\eta = \gamma = \sqrt{\frac{L\ln(L|X||A|/\delta)}{T|X||A|}}$, guarantees that with probability at least $1 - 19\delta$:*

$$\alpha\text{-}R_T \leq 14L|X|^2\sqrt{2T|A|\ln\left(\frac{T|X|^2|A|}{\delta}\right)}$$

*and*

$$V_t \leq 61L|X|\sqrt{2t|A|\ln\left(\frac{2mT^2|X||A|}{\delta}\right)}$$

*for all $t \in [T]$, where $\alpha = \frac{\rho}{1+\rho}$.*

*Proof.* It is sufficient to combine Theorem B.2 and Theorem 4.5. Specifically, with probability at least $1 - 15\delta$, for all $\tilde{q} \in \Delta^\diamond \subset \widehat{\Delta}_t(\mathcal{P}_t)$, we have:

$$\sum_{t \in [T]} r_t^\top (\tilde{q} - q_t) \leq 14L|X|^2\sqrt{2T|A|\ln\left(\frac{T|X|^2|A|}{\delta}\right)}.$$

Let $q^\dagger = \arg\max_{q \in \Delta(M)} \sum_{t=1}^T r_t^\top q$. We observe that:

$$\bar{q} = \frac{L}{L+\rho'}q^\diamond + \frac{\rho'}{L+\rho'}q^\dagger \in \Delta^\diamond.$$

Thus, it holds:

$$\sum_{t=1}^T r_t^\top \bar{q} = \sum_{t=1}^T r_t^\top \left(\frac{L}{L+\rho'}q^\diamond + \frac{\rho'}{L+\rho'}q^\dagger\right) \geq \frac{\rho'}{L+\rho'}\sum_{t=0}^T r_t^\top q^\dagger.$$

This proves that with probability at least $1 - 15\delta$:

$$\left(\frac{\rho'}{L + \rho'}\right) \text{-} R_T \leq 14L|X|^2 \sqrt{2T|A| \ln\left(\frac{T|X|^2|A|}{\delta}\right)}.$$

Similarly to the stochastic case, employing Theorem B.7, with probability at least $1 - 4\delta$, we have:

$$V_t \leq 61L|X| \sqrt{2t|A| \ln\left(\frac{2mT^2|X||A|}{\delta}\right)}.$$

By Union Bound this holds with probability $1 - 19\delta$.

Noticing that $\frac{\rho}{1+\rho'} = \frac{\rho'}{L+\rho}$ concludes the proof. □

### B.8 Positive Violation Bound

In this section, we provide the results on the positive violation bound attained by Algorithm 1.

**Theorem 4.4.** *Let $\delta \in (0,1)$. In the stochastic setting, Algorithm 1 guarantees with probability at least $1 - 16\delta$:*

$$\mathcal{V}_t \leq 18L|X| \sqrt{2t|A| \ln \frac{2mT|X||A|}{\delta}}, \quad \forall t \in [T].$$

*Proof.* Define for each $i \in [m]$ and $t \in [T]$ the following quantity:

$$\mathcal{V}_{t,i} := \sum_{\tau=1}^{t} \left[\bar{g}_i^\top q_\tau\right]^+.$$

Given an $i \in [m]$ and a $t \in [T]$ we have:

$$\mathcal{V}_{t,i} = \sum_{\tau=1}^{t} \left[\bar{g}_i^\top q_\tau\right]^+$$

$$= \sum_{\tau=1}^{t} \left[(\bar{g}_i - \widehat{g}_{\tau,i} + \widehat{g}_{\tau,i})^\top q_\tau\right]^+$$

$$= \sum_{\tau=1}^{t} \left[(\bar{g}_i - \widehat{g}_{\tau,i})^\top q_\tau + \widehat{g}_{\tau,i}^\top q_\tau\right]^+$$

$$= \sum_{\tau=1}^{t} \left[(\bar{g}_i - \widehat{g}_{\tau,i})^\top q_\tau + \widehat{g}_{\tau,i}^\top q_\tau - \widehat{g}_{\tau,i}^\top \widehat{q}_\tau + \widehat{g}_{\tau,i}^\top \widehat{q}_\tau\right]^+$$

$$\leq \sum_{\tau=1}^{t} \left[(\bar{g}_i - \widehat{g}_{\tau,i})^\top q_\tau + \widehat{g}_{\tau,i}^\top q_\tau - \widehat{g}_{\tau,i}^\top \widehat{q}_\tau + b_\tau^\top \widehat{q}_\tau\right]^+ \tag{34}$$

$$\leq \sum_{\tau=1}^{t} \left[(\bar{g}_i - \widehat{g}_{\tau,i})^\top q_\tau + b_\tau^\top \widehat{q}_\tau\right]^+ + \|q_\tau - \widehat{q}_\tau\|_1$$

$$\leq \sum_{\tau=1}^{t} \left[(\bar{g}_i - \widehat{g}_{\tau,i})^\top q_\tau + b_\tau^\top \widehat{q}_\tau\right]^+ + 2L|X|\sqrt{2t \ln \frac{2L}{\delta}} + 3L|X|\sqrt{2t|A| \ln \frac{2T|X||A|}{\delta}} \tag{35}$$

$$\leq \sum_{\tau=1}^{t} \left[b_\tau^\top q_\tau + b_\tau^\top \widehat{q}_\tau\right]^+ + 5L|X|\sqrt{2t|A| \ln \frac{2T|X||A|}{\delta}}, \tag{36}$$

where Inequality (34) holds since $\widehat{q}_\tau \in \widehat{\Delta}_t(\mathcal{P}_t)$, Inequality (35) follows from Lemma B.3 of (Rosenberg & Mansour, 2019b) with probability at least $1 - 2\delta$ and Inequality (36) holds by Lemma 4.1 with probability $1 - 11\delta$ jointly for each $i$ and $t$.

Since $b_t = \sqrt{\frac{2\ln\left(\frac{2m|X||A|T}{\delta}\right)}{N_t(x,a)}}$ by Lemma B.6 with probability at least $1 - 3\delta$, employing a Union Bound we have, with probability at least $1 - 16\delta$:

$$\mathcal{V}_{t,i} \leq 2L\sqrt{2T\ln\frac{1}{\delta}} + 4\sqrt{2|X||A|Lt\ln\left(\frac{2mT|X||A|}{\delta}\right)} + 12L|X|\sqrt{2t|A|\ln\frac{2mT|X||A|}{\delta}}$$

$$\leq (2 + 4 + 12)L|X|\sqrt{2t|A|\ln\frac{2mT|X||A|}{\delta}}$$

$$= 18L|X|\sqrt{2t|A|\ln\frac{2mT|X||A|}{\delta}},$$

for all $i \in [m], t \in [T]$. This concludes the proof. □

## C   TECHNICAL LEMMAS

In this section we provide some auxiliary lemmas which are needed throughout the paper.

We start by the following application of the Hoeffding inequality on the constraints.

**Lemma C.1.** *Let $\delta \in (0,1)$. With probability at least $1 - \delta$ it holds, for all $(x,a) \in X \times A$, $i \in [m]$, $t \in [T]$:*

$$\left| \frac{1}{N_t(x,a)} \sum_{\tau \in \mathcal{T}_{t-1,x,a}} g_{\tau,i}(x,a) - \bar{g}_i(x,a) \right| \leq \sqrt{\frac{2\ln\frac{2m|X||A|T}{\delta}}{N_t(x,a)}}.$$

*Proof.* The proof is a simple application of Hoeffding's inequality and a union bound. □

We proceed with a concentration result on the transition functions.

**Lemma C.2** (Lemma J.6 of (Stradi et al., 2025c)). *For any $\delta \in (0,1)$, let $\{\pi_t\}_{t=1}^T$ be policies, then for any collection of transition $P_t^x \in \mathcal{P}_t$ with probability at least $1 - 2\delta$, it holds:*

$$\sum_{t=1}^T \|q^{P,\pi_t} - q^{P_t^x,\pi_t}\|_1 \leq 2L|X|^2\sqrt{2T\ln\left(\frac{L|X|}{\delta}\right)} + 3L|X|^2\sqrt{2T|A|\ln\left(\frac{T|X|^2|A|}{\delta}\right)}.$$

Thus, we provide concentration results for the constraints.

**Lemma C.3.** *For any $\delta \in (0,1)$, let $f_t : X \times A \to [-1,1]$ be a sequence of functions that is $t-1$ predictable, and let $\pi_t$ be a randomized policy. Then, with probability at least $1 - \delta$, it holds:*

$$\left| \sum_{t \in [T]} \sum_{x \in X, a \in A} f_t(x,a)\mathbb{I}_t\{x,a\} - \sum_{t \in [T]} f_t^\top q^{P,\pi_t} \right| \leq 2L\sqrt{2T\ln\frac{1}{\delta}},$$

*where $\mathbb{I}_t\{x,a\} = 1$ if and only if the pair $(x,a)$ is visited in episode $t$.*

*Proof.* By definition of the occupancy measure, it holds:

$$\mathbb{E}\left[f_t(x,a)\mathbb{I}_t\{x,a\}|P,\pi_t\right] = \sum_{x \in X} \sum_{a \in A} q_t(x,a)f_t(x,a) = f_t^\top q_t.$$

We defined the following sequence:

$$X_t = \sum_{\tau=1}^t \left[ \sum_{x \in X, a \in A} f_\tau(x,a)\mathbb{I}_\tau\{x,a\} - f_\tau^\top q^{P,\pi_\tau} \right].$$

$X_t$ is a Martingale difference sequence and $|X_t - X_{t-1}| \le 2L$. Applying the Azuma inequality, we obtain that with probability at least $1 - \delta$:

$$\left| \sum_{t \in [T]} \sum_{x \in X, a \in A} f_t(x, a) \mathbb{I}_t\{x, a\} - \sum_{t \in [T]} f_t^\top q^{P, \pi_t} \right| \le 2L \sqrt{2T \ln \frac{1}{\delta}}.$$

This concludes the proof. □

**Lemma C.4.** *For any $\delta \in (0, 1)$, for any sequence of occupancy measure $\bar{q}_t \in \widehat{\Delta}_t(\mathcal{P}_t)$ and any function $f_t(x, a)$ sampled from a distribution with mean $\bar{f}(x, a)$, i.e., $\mathbb{E}[f_t(x, a)] = \bar{f}(x, a)$ and $\mathbb{P}(|f_t(x, a)| \le 1) = 1$, it holds that with probability at least $1 - \delta$:*

$$\left| \sum_{t \in [T]} \bar{f}^\top \bar{q}_t - \sum_{t \in [T]} f_t^\top \bar{q}_t \right| \le 2L \sqrt{2T \ln \frac{1}{\delta}}.$$

*Proof.* The proof follows the one of Lemma C.3, after noticing that the quantity of interest is a Martingale difference sequence. □

Thus, we provide an auxiliary result on the concentration of the optimistic loss estimator.

**Lemma C.5.** *For any $\delta \in (0, 1)$, Algorithm 1 attains, with probability at least $1 - 7\delta$:*

$$\sum_{t=1}^T (\ell_t - \widehat{\ell}_t)^\top \widehat{q}_t \le \gamma |X| |A| T + 2L |X|^2 \sqrt{2T \ln \left( \frac{L|X|}{\delta} \right)} + 3L |X|^2 \sqrt{2T |A| \ln \left( \frac{T|X|^2 |A|}{\delta} \right)}$$

*Proof.* The result follows from the proof of Lemma 6 from (Jin et al., 2020) and employing Lemma C.2. □

## D    ADDITIONAL EXPERIMENTS

In this section, we describe the experiments that show the theoretical guarantees of our algorithm in practice. Our goal is to assess WC-OPS's performance in both stochastic and adversarial setting and to compare it with state-of-the-art algorithms from literature. Specifically, the algorithms we consider are:

- OptCMDP (Algorithm 1 of (Efroni et al., 2020)). This algorithm solves an optimistic linear programming formulation of the CMDP, for each episode. OptCMDP attains $\widetilde{\mathcal{O}}(\sqrt{T})$ regret and *positive* violation, without Slater's condition, being arguably state-of-the-art in terms of performance for the stochastic setting.

- OptPrimalDual-CMDP (Algorithm 4 of (Efroni et al., 2020)). This algorithm employs a primal-dual approach, performing incremental updates for both the primal (that is, the policy) and dual Lagrange variables. OptPrimalDual-CMDP attains $\widetilde{\mathcal{O}}(\frac{1}{\rho}\sqrt{T})$ regret and violation, assuming Slater's condition.

- Greedy, a greedy-like algorithm which employs the empirical average for every estimate. It works similarly to OptCMDP, without relying on confidence intervals.

All experiments are conducted in a finite-horizon CMDP with a layered structure. To achieve a fair comparison, we have aligned all the algorithms to our environment; in particular, we make the following assumptions:

- The CMDP has a layered structure and is loop-free. The "length" of the episodes $H$ of (Efroni et al., 2020) corresponds in our setting to the number of layers $L$.

- Each layer has its own states and the first and last layer only contain one state, differently from (Efroni et al., 2020) where set of states that can be visited by the algorithm is the same at each episode's step.

- The episode always starts from layer 0, thus there is no need to keep the initial state distribution $\mu$ mentioned in (Efroni et al., 2020).

- Rewards are in $[0, 1]$ and constraints are in $[-1, 1]$ for each $i = 1, ..., m$, differently from (Efroni et al., 2020), where constraints are non-negative.

The parameter $\delta$ is set to $0.01$ for all experiments. In the stochastic setting, the values of reward and constraints are sampled from a Bernoulli distribution (rescaled to $[-1, 1]$ in the case of constraints). In the adversarial setting, reward and constraints are generated by an OGD algorithm (Orabona, 2019) which receives as a gradient a vector containing for each state the negative product of the policy played at that round and a fixed initial vector of rewards (or constraints).

To obtain statistically robust results, each experiment is repeated a certain number of times ($n \simeq 10$). The runs are executed in parallel using a process pool to reduce computational time. For each algorithm, we report the average performance together with $95\%$ confidence intervals.

The experiments were conducted in three different settings:

- With stochastic reward and stochastic constraints, we compared our algorithm with `OptCMDP` and `OptPrimalDual-CMDP`.

- With adversarial reward and stochastic constraints, we compared our algorithm with `OptCMDP` and `Greedy`.

- With adversarial reward and adversarial constraints, we compared our algorithm with `Greedy`.

`OptCMDP` and `OptPrimalDual-CMDP` both attain $\widetilde{\mathcal{O}}(\sqrt{T})$ regret and violation in the stochastic setting. `Greedy` has no guarantees of sublinearity. In the stochastic case, we expect our algorithm to perform similarly to `OptCMDP`, which is taylored to this setting, and better than `OptPrimalDual-CMDP`. In the adversarial case we expect `WC-OPS` to outperform `Greedy`.

### D.1 STOCHASTIC REWARD AND STOCHASTIC CONSTRAINTS

In this section, we provide the experiments in the fully stochastic environment.

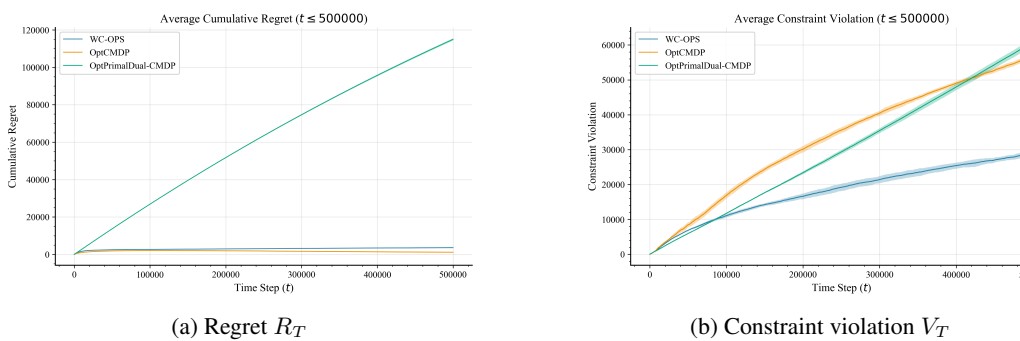

(a) Regret $R_T$             (b) Constraint violation $V_T$

Figure 3: Stochastic reward and stochastic constraints.

In Figures 3a - 3b, we provide the experiments presented in the main paper. In Figures 4a - 4b, we provide a novel experiment. As in the previous one, `WC-OPS` achieves a performance similar to the one of `OptCMDP` and better with respect to `OptPrimalDual-CMDP` in terms of regret. The violation performance is similar across the algorithms. Finally, in Figures 5a - 5b, we show the results from a final experiment in the stochastic setting.

### D.2 ADVERSARIAL REWARD AND STOCHASTIC CONSTRAINTS

In this section, we provide the experiments when the environment encompasses adversarial rewards and stochastic constraints.

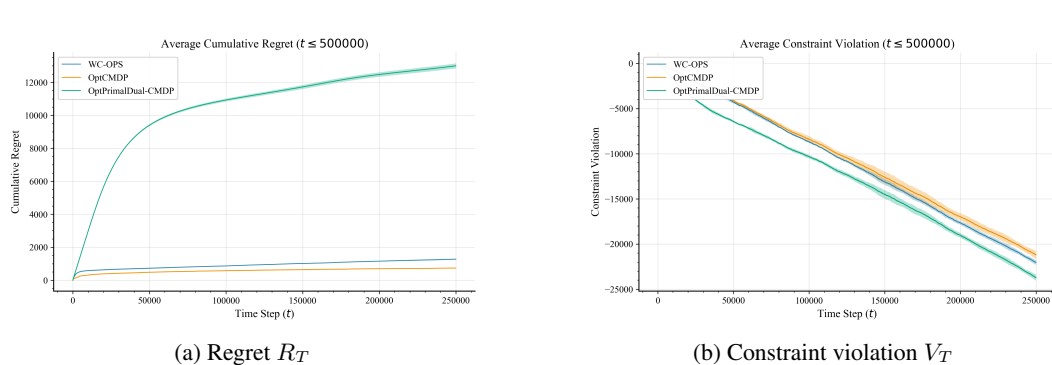

(a) Regret $R_T$

(b) Constraint violation $V_T$

Figure 4: Stochastic reward and stochastic constraints.

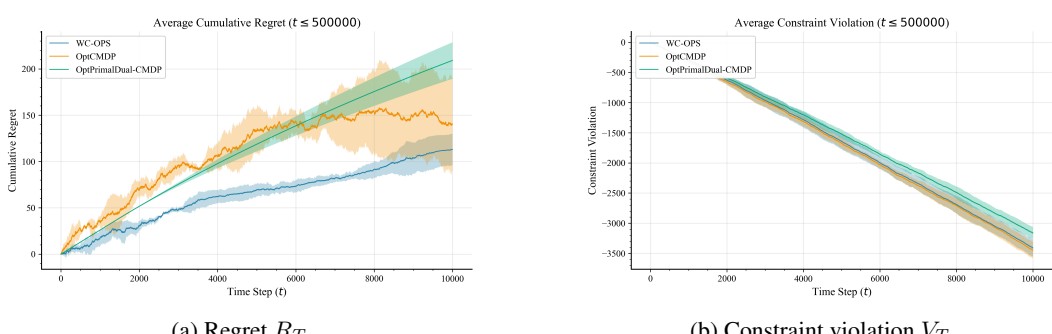

(a) Regret $R_T$

(b) Constraint violation $V_T$

Figure 5: Stochastic reward and stochastic constraints.

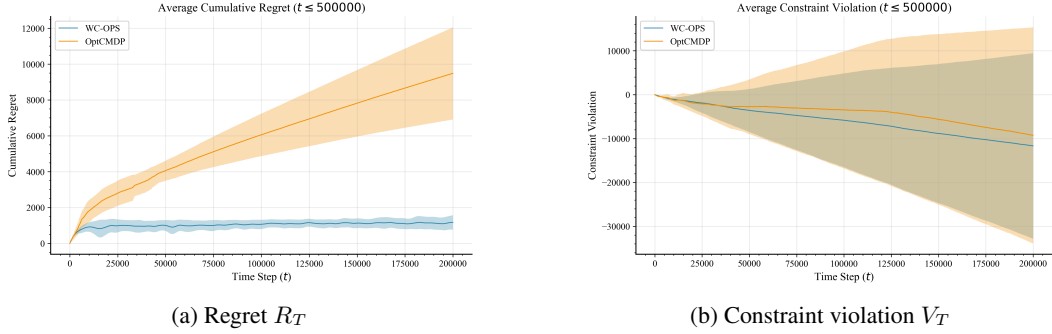

(a) Regret $R_T$

(b) Constraint violation $V_T$

Figure 6: Adversarial reward and stochastic constraints.

In Figures 6a - 6b, we provide our first experiment for the setting. As expected, `WC-OPS` outperforms in terms of regret `OptCMDP`, while attaining similar constraints violation guarantees. Finally, Figures 7a - 7b provide a similar experiment.

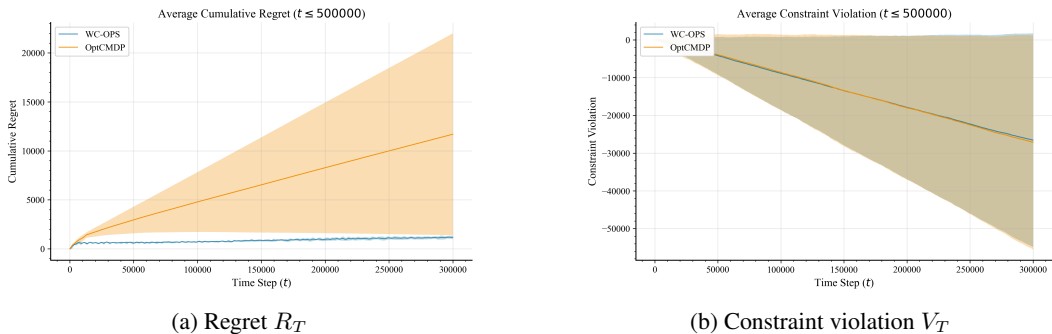

(a) Regret $R_T$             (b) Constraint violation $V_T$

Figure 7: Adversarial reward and stochastic constraints.

### D.3 ADVERSARIAL REWARD AND ADVERSARIAL CONSTRAINTS

In this section, we provide the experiments when the environment is completely adversarial.

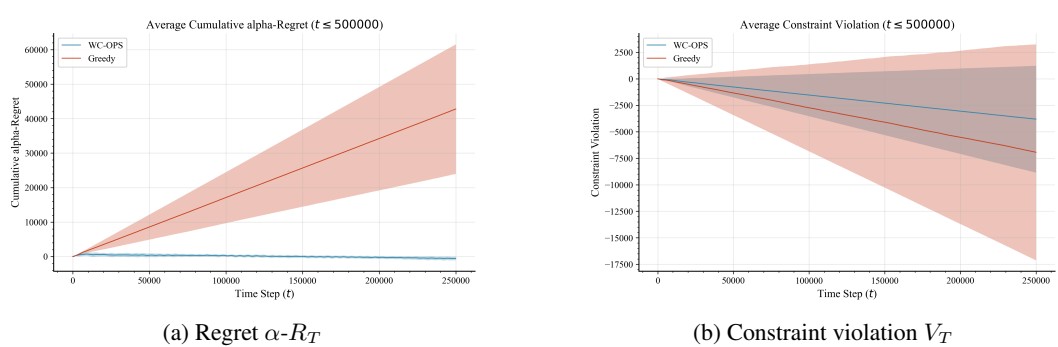

(a) Regret $\alpha$-$R_T$             (b) Constraint violation $V_T$

Figure 8: Adversarial reward and adversarial constraints.

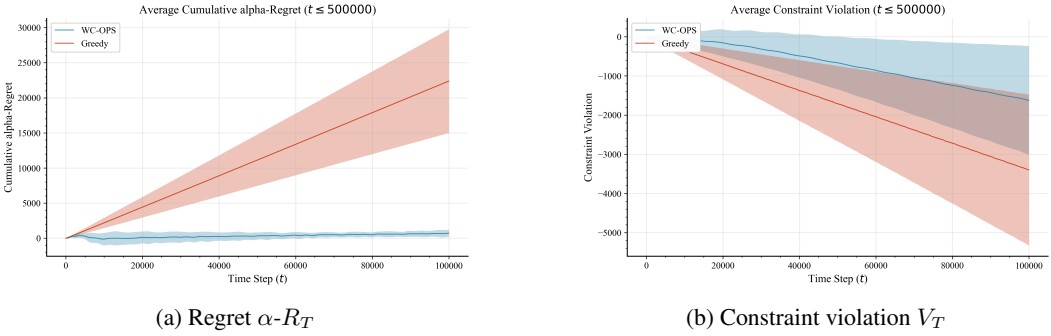

(a) Regret $\alpha$-$R_T$             (b) Constraint violation $V_T$

Figure 9: Adversarial reward and adversarial constraints.

In Figures 8a - 8b, we provide the first experiment in the adversarial setting. As expected, `WC-OPS` significantly outperforms `Greedy` in terms of $\alpha$-Regret, while, in this case, attains a similar performance in terms of violation. Finally, in Figures 9a - 9b, we provide a similar experiment.

## D.4 LEARNING DYNAMICS IN THE SIMPLEX

In this section, we provide some graphical representations of the dynamics of Algorithm 1. The experiments are conducted in a single state environment with three actions. Specifically, in Figures 10 - 11 - 12, it is possible to verify that Algorithm 1 converges asymptotically to the true decision space.

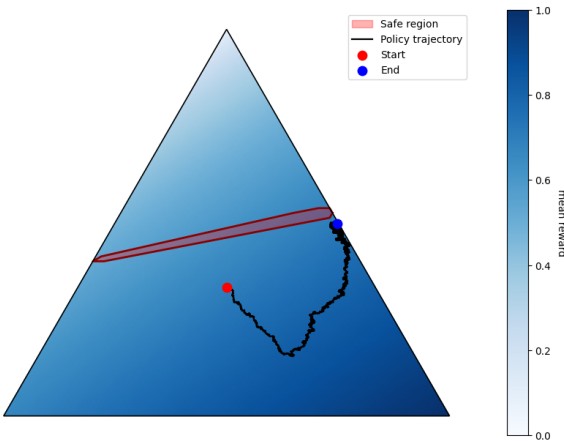

Figure 10: Learning dynamics of Algorithm 1

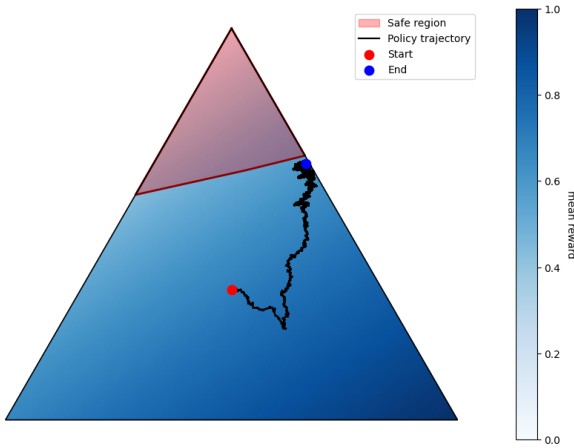

Figure 11: Learning dynamics of Algorithm 1

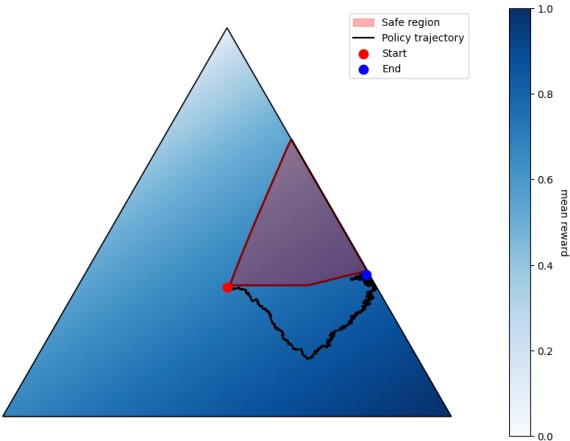

Figure 12: Learning dynamics of Algorithm 1

# LLM USAGE

We used LLMs for polishing the writing, only.

