# OpenReview forum: "Beyond Slater’s Condition in Online CMDPs with Stochastic and Adversarial Constraints"
_ICLR.cc/2026/Conference — Submitted to ICLR 2026_

### Official Review · Reviewer_5J13 · 2025-10-30

**Soundness:** 2
**Presentation:** 3
**Contribution:** 2
**Rating:** 4
**Confidence:** 3

**Summary:**

The paper studies online CMDPs where rewards may be adversarial, and constraints are either stochastic or adversarial. It proposes WC-OPS (Weighted Constrained Optimistic Policy Search), a constrained optimization-based method that performs online mirror descent over a moving, optimism-based feasible set. The authors prove, without requiring Slater’s condition, $O(\sqrt{T})$ regret and positive violation in the stochastic constraints case; in the adversarial constraints case they obtain sublinear violation and sublinear $\alpha$-regret against the unconstrained optimum.

**Strengths:**

This paper establishes best-of-both-worlds guarantees for online CMDPs with stochastic and adversarial constraints; in the stochastic setting, it further provides positive-violation guarantees and shows Slater’s condition is not necessary.

**Weaknesses:**

- Although the paper improves upon Stradi et al. (2025c) by both tightening the guarantees and eliminating reliance on Slater’s condition, these gains are obtained by computing a safe decision set and solving a constrained optimization each round, incurring higher computational overhead than the primal–dual method in Stradi et al. (2025c). Because the proofs largely defer to the correctness of the safe set estimation, the analysis is comparatively straightforward; in effect, the paper’s main contribution is the weighting strategy for constraint estimation.

- The ``Slater-free'' statement is potentially misleading. In the adversarial regime, the analysis requires a Slater-like margin to obtain $\alpha$-regret, arguably a stricter variant of Slater’s condition than the classical form.

- The paper repeatedly contrasts its theory with Stradi et al. (2025c), but the experiments do not include that baseline. The submission also lacks a systematic treatment of computational complexity.

- The paper is largely built on the paper on constrained bandits, "Beyond primal-dual methods in bandits with stochastic and adversarial constraints," and extends it into the CMDP setting; most of the techniques are from this paper. It is unclear what the technical challenges and contributions are.

- The paper focused on model-based design and optimization on a stationary measure space. It might be infeasible in a large state-action space, and it would be better to consider the policy optimization.

**Questions:**

Please see the weakness above.

---

> ### Author Response · Authors · 2025-11-14
>
> We thank the Reviewer for the effort in evaluating our work.
>
> > On the higher computational overhead w.r.t. the primal–dual method in Stradi et al. (2025c).
>
>  We agree with the Reviewer that Stradi et al. (2025c) avoid the computational overhead of solving the projection step. Nonetheless, we underline the following key aspects. First, the primal algorithm in Stradi et al. (2025c) needs to compute the optimistic bonus term to build the estimate of the "Q-function". This step can be done algorithmically but has a computational complexity of order $O(|X|^3|A|^2)$. Second, it is standard in the adversarial MDPs literature to build algorithms that have to solve projection steps at each episode (e.g., [1],[2],[3]).
>
> [1] "Learning Adversarial MDPs with Bandit Feedback and Unknown Transition", ICML 2020
>
> [2] "Online stochastic shortest path with bandit feedback and unknown transition function", NeurIPS 2019
>
> [3] "Online convex optimization in adversarial Markov decision processes", ICML 2019
>
> > Because the proofs largely defer to the correctness of the safe set estimation, the analysis is comparatively straightforward; in effect, the paper’s main contribution is the weighting strategy for constraint estimation.
>
> We see it as a point in favor of our work, since we believe that this is a consequence of the natural and intuitive structure of our algorithm, which directly translates into a simpler analysis. In contrast, primal-dual methods often rely on less intuitive constructions and require more elaborate proofs (for instance, to bound the behavior of the Lagrangian variables) while leading to **weaker** theoretical guarantees.
>
> > The ``Slater-free'' statement is potentially misleading.
>
> We thank the Reviewer for the opportunity to clarify this aspect. Our algorithm effectively goes beyond Slater's condition on multiple aspects. Indeed, in the stochastic regime, we neither assume Slater nor have the $1/\rho$ dependence in the regret/violation bounds. In the adversarial setting, the same reasoning holds for the violation. For the regret, the Reviewer is correct in stating that Slater's is assumed to compute the $\alpha$-regret. Nonetheless, the $\alpha$-regret is the metric employed to show best-of-both-worlds guarantees in the literature (e.g., [4],[5],[6]), since it is provably not possible to obtain sublinear violation and sublinear $\alpha$-regret with a better notion of $\alpha$ (see Theorem 4.1 of [7]). Remarkably, even in this case, the $\alpha$-regret bound does not have any dependence on $1/\rho$, thus improving previous results for CMDPs.
>
> We will surely clarify this point in the final version of the paper.
>
> [4] "A Unifying Framework for Online Optimization with Long-Term Constraints", NeurIPS 2022
>
> [5] "Policy Optimization for CMDPs with Bandit Feedback: Learning Stochastic and Adversarial Constraints", ICML 2025
>
> [6] "The Best of Many Worlds: Dual Mirror Descent for Online Allocation Problems", Operations Research
>
> [7] "No-regret is not enough! bandits with general constraints through adaptive regret minimization", ICML 2025
>
> > The paper repeatedly contrasts its theory with Stradi et al. (2025c), but the experiments do not include that baseline.
>
> The choice of focusing mainly on stochastic settings in the main paper is due to the fact that best-of-both-worlds algorithms tend to perform worse in stochastic settings. Indeed, we wanted to compare our algorithm with the strongest benchmark in the literature for stochastic CMDPs, that is, Algorithm 1 of [Efroni et al., 2020], which, as in our case, does not require Slater's condition to hold and does not have any $1/\rho$ dependency on the regret bound.
>
> We commit to inserting in the final version of the paper experiments against the algorithm of [Stradi et al. (2025c)] with an additional comparison in terms of empirical computational costs.
>
> > It is unclear what the technical challenges and contributions are.
>
> We agree with the Reviewer that our techniques extend similar results for the easier MAB setting. However, the extension to CMDP settings introduces technical challenges that prevent us from employing such results as a black box. For instance, one challenge is that the transitions are not known, and thus the moving decision space also depends on the uncertainty on the transitions. We believe that extending such techniques in order to considerably improve the state-of-the-art on CMDPs should be evaluated positively.
>
> > On the extension to policy optimization.
>
> We agree with the Reviewer that extending our results to model-free settings is an interesting future direction. Nonetheless, we underline that it is not possible to project the policy in the safe decision space optimizing state by state, as in model-free algorithms. Thus, we believe that it would require a complete rethinking of our approach.
>
> We hope to have addressed the Reviewer's concerns. Please let us know if further discussion is necessary.

---

### Official Review · Reviewer_bn5r · 2025-10-30

**Soundness:** 3
**Presentation:** 3
**Contribution:** 3
**Rating:** 6
**Confidence:** 3

**Summary:**

This paper studies online episodic constrained Markov decision processes (CMDPs) under both stochastic and adversarial settings and proposes a new algorithm, Weighted Constrained Optimistic Policy Search (WC-OPS). WC-OPS departs from standard primal–dual methods by performing no-regret updates via online mirror descent over a moving decision space that adapts to online constraint estimates, coupled with a UCB-like optimism mechanism. In the stochastic setting, the algorithm attains $\tilde{O}(\sqrt{T})$ regret and $\tilde{O}(\sqrt{T})$ cumulative constraint violation without requiring Slater’s condition. In the adversarial setting, it achieves sublinear cumulative constraint violation without Slater’s condition and sublinear $\alpha$-regret relative to the unconstrained optimum. Numerical experiments indicate that WC-OPS delivers competitive performance compared to state-of-the-art baselines.

**Strengths:**

1. This paper introduces a weighted constraint estimation scheme with a moving feasible set instead of the standard primal-dual framework, which is more adaptive and avoids the discussion about Lagrangian relaxations. The authors also introduce a positive violation metric that prevents compensation across episodes, which is more practical and stringent.

2. This paper achieves optimal $O(\sqrt{T})$ worst-case regret in the stochastic setting without Slater’s condition, improving the results of prior work which required it or achieved worse rates $O(T^{−3/4})$.

3. The paper provides high-probability regret and violation bounds with detailed proofs in the appendix.

4. The paper is well-structured with a clear statement of analysis and detailed explanation of algorithm, table and figures.

**Weaknesses:**

1. This algorithm only serves for the tabular setting, which weakens its practicality and versatility in real-world problems.

2. The algorithm needs to maximize the occupancy measure over a transition confidence set, which could be time-consuming and have computational cost. It would be better to discuss the computational cost.

3. The definition of the problem-specific parameter $\rho$ seems to be too complex. It may be meaningless or difficult to fetch when applying this algorithm to real problems.

4. The paper only conducts experiments in synthetic, layered MDPs. The performance
of the algorithm under the real-world environment remains unknown.

**Questions:**

1. How sensitive is the algorithm to the choice of parameters $\eta$, $\gamma$ and $\delta$? Can the authors improve the
results by tuning the parameters?

2. What are the pros and cons of WC-OPS compared to primal-dual method? Can the authors provide a more detailed clarification for choosing WC-OPS?

3. The authors define $\alpha$-regret with respect to the unconstrained optimum in adversarial setting. Why do the authors choose this baseline instead of the most common optimum baseline?

---

> ### Author Response · Authors · 2025-11-14
>
> We thank the Reviewer for the positive evaluation of our work.
>
> > How sensitive is the algorithm to the choice of parameters $\eta$, $\gamma$ and $\delta$? Can the authors improve the results by tuning the parameters?
>
> In the experiments, $\eta$ and $\gamma$ were set following their theoretical prescriptions (Appendix B.1) to ensure the validity of the $\widetilde{\mathcal{O}}(\sqrt{T})$ bounds. As for $\delta$, it was set to small values (0.01, 0.015). The algorithm is robust against a wide range of values: $\eta$ and $\gamma$ mainly affect convergence speed, while $\delta$ influences the concentration confidence and must be set sufficiently small.
>
> > What are the pros and cons of WC-OPS compared to primal-dual method? Can the authors provide a more detailed clarification for choosing WC-OPS?
>
> WC-OPS provides a completely different framework with respect to primal-dual methods, directly operating on a dynamic decision space and without introducing dual variables. As we prove, this approach yields stronger guarantees in terms of regret and violation. In the following, we highlight the comparison with the primal-dual algorithm provided in [Stradi et al. 2025c], that is, the state-of-the-art primal-dual method for CMDPs where the constraints can be either stochastic or adversarial. In the stochastic setting, WC-OPS attains a $\widetilde{\mathcal{O}}(\sqrt{T})$ bound for the regret and the violation without relying on Slater's condition, while Stradi et al. (2025c) achieve $\widetilde{\mathcal{O}}(\min(\frac{1}{\rho^2}\sqrt{T}, T^{\frac{3}{4}} ))$. We additionally provide a $\widetilde{\mathcal{O}}(\sqrt{T})$ bound for the **positive** constraint violation, a stronger metric preventing cancellations among terms. In the adversarial setting, WC-OPS provides $\widetilde{\mathcal{O}}(\sqrt{T})$ violation without requiring Slater's condition, while Stradi et al. (2025c) provide $\widetilde{\mathcal{O}}(\frac{1}{\rho^2}\sqrt{T})$, under Slater's condition. Finally, we obtain a $\widetilde{\mathcal{O}}(\sqrt{T})$ bound with respect to the unconstrained optimum instead of the $\widetilde{\mathcal{O}}(\frac{1}{\rho^2}\sqrt{T})$ bound with respect to the constrained one of Stradi et al. (2025c).
>
> > The authors define $\alpha$-regret with respect to the unconstrained optimum in adversarial setting. Why do the authors choose this baseline instead of the most common optimum baseline?
>
> If the Reviewer is referring to the choice of employing the unconstrained optimum in the $\alpha$-regret definition, we remark that the unconstrained optimum is a stronger baseline with respect to the constrained one. Thus, our guarantees against the unconstrained optimum immediately imply the same guarantees against the constrained one. This choice highlights that WC-OPS achieves sublinear performance even under the most demanding benchmark.
>
> Differently, if the Reviewer is referring to the choice of employing the $\alpha$-regret metric instead of the standard regret, this is done since it is provably not possible to attain both sublinear regret and sublinear violation when the constraints are adversarial ([1]). We remark that this choice is standard in the literature (e.g., [2],[3],[4]) and necessary (see Theorem 4.1 of [5]).
>
> Please let us know if further discussion is necessary.
>
> [1] "Online learning with sample path constraints", JMLR 2009
>
> [2] "A Unifying Framework for Online Optimization with Long-Term Constraints", NeurIPS 2022
>
> [3] "Policy Optimization for CMDPs with Bandit Feedback: Learning Stochastic and Adversarial Constraints", ICML 2025
>
> [4] "The Best of Many Worlds: Dual Mirror Descent for Online Allocation Problems", Operations Research
>
> [5] "No-regret is not enough! bandits with general constraints through adaptive regret minimization", ICML 2025

---

### Official Review · Reviewer_4xhw · 2025-10-31

**Soundness:** 2
**Presentation:** 2
**Contribution:** 2
**Rating:** 2
**Confidence:** 4

**Summary:**

The paper studies online episodic CMDPs with bandit feedback, unknown transitions, adversarial rewards, and either stochastic or adversarial constraints. It proposes WC OPS, an optimistic occupancy measure algorithm with a moving feasible set built from transition confidence sets and optimism bonuses on constraints, updated by an online mirror descent step with implicit exploration. Claimed rates: in the stochastic constraints case, $O(\sqrt{T})$ regret, standard violation, and positive violation, all without Slater; in the adversarial constraints case, sublinear violation without Slater and sublinear $\alpha$ regret against the unconstrained optimum with $\alpha=\rho/(1+\rho)$. Synthetic experiments compare against OptCMDP and a primal dual baseline.

The results for the stochastic setting are standard and not novel, even without assuming Slater’s condition. The adversarial setting adds little new insight—its $\alpha$-regret formulation restates known impossibility results and introduces a weaker comparator defined by $\alpha=\rho/(1+\rho)$, which never recovers standard regret even for benign constraint sequences. Overall, I think the paper’s “best-of-both-worlds” claim is overstated: the moving-weights mechanism is reasonable but conventional, while the theoretical and empirical results offer limited genuine advancement.

**Strengths:**

First, removing Slater in the stochastic constraints case while keeping \sqrt{T} rates for regret, violation, and positive violation is an attractive message for safety-sensitive learning.

The moving feasible set with an optimism bonus on constraints is a clean alternative to primal-dual updates. It aligns with known optimistic design for losses and transitions, and it avoids tuning of dual steps tied to Slater.

**Weaknesses:**

Computing $u_t(s,a)$ over a confidence set of transitions can be very expensive, since it is an occupancy maximization over a polytope of MDPs. The paper does not discuss how this maximization is performed or approximated in practice per episode. Any complexity statement or an efficient reduction would help.


The experiments are limited to synthetic settings with stochastic constraints. There is no evaluation under adversarial constraints and no test on larger CMDPs or tasks with richer state spaces. Since the claims include both regimes, at least one adversarial constraints experiment would be helpful, and some wall clock results would show the cost of computing confidence sets, optimistic occupancies, and the OMD projection.


In the adversarial constraints case, the move to cumulative alpha regret is standard, but the definition of alpha depends on a margin rho that is the worst slack over all constraints and all state action pairs in the support of a strictly feasible occupancy. In realistic CMDPs, rho can be very small, which makes alpha near zero and the regret target nearly vacuous. The paper should either quantify rho for natural benchmark families or provide a lower bound on rho under standard reachability or controllability assumptions. As it stands, the claim of sublinear alpha regret can hide a very weak comparator.

The adaptive term $\Gamma_{t,i}$ is presented as the mechanism that allows the same algorithm to handle both stochastic and adversarial settings, automatically adjusting the learning rate so that when violations are small ($\Gamma_{t,i}=0$) it behaves like the stochastic estimator, and when violations are large it adapts to adversarial feedback. However, the theoretical separation between these two modes is not rigorously justified.
1. The $\alpha$-regret analysis in the adversarial regime depends on a problem-specific margin $\rho$ and defines $\alpha = \rho / (1+\rho)$. This margin also governs the adaptive term indirectly, but the algorithm never estimates or uses $\rho$ explicitly. As a result, the paper implies that the same procedure works for both regimes ``without knowing the setting,'' yet it does not show that $\Gamma_{t,i}$ correctly tracks or bounds $\rho$ in mixed or intermediate regimes.
2. What if in the adversarial setting, the rewards are chosen as stochastic but constraints are adversarial, the analysis offers no guidance on which comparator applies. Since the impossibility result arises from adversarial constraints, even mild adversariality should still yield only $\alpha$-regret, not standard regret. The paper does not articulate this distinction, leaving unclear whether such mixed settings revert to standard regret or remain constrained to $\alpha$-regret bounds.
3. If the adversary chooses constraints that always satisfy the feasibility condition with large uniform slack, then theoretically $\rho$ could be large and $\alpha \rightarrow 1$.  The algorithm should then reduce to the unconstrained (or stochastic) case. Yet the results never tighten to full-regret bounds as $\alpha \to 1$; the analysis remains fixed at the weaker $\alpha$-regret level. This gap means the method cannot recognize when the problem effectively becomes unconstrained or easy.

Because of these points, I believe the claim of being ``best-of-both-worlds'' is **overstated.** The algorithm indeed shares one form across regimes, but its guarantees remain conservative (\emph{$\alpha$-regret}) whenever constraints are adversarial, even if they are benign.

**Questions:**

Could you explain which technical step specifically avoids the dependence on a strict feasibility margin, and how the circular reasoning around $\Gamma_{t,i}=0$ is resolved rigorously?

Can you formally show that $\Gamma_{t,i}$ remains small (or zero) in the stochastic case and stabilizes in the adversarial case without knowing the regime a priori?

Since $\rho$ can be arbitrarily small, doesn’t this make $\alpha$-regret potentially vacuous?
Can you show any setting where $\alpha$ is meaningfully bounded away from zero?

What is the per-episode computational complexity, and how is $u_t$ computed in your experiments

---

> ### Author Response · Authors · 2025-11-14
>
> We thank the Reviewer for the effort in evaluating our work.
>
> > Could you explain which technical step specifically avoids the dependence on a strict feasibility margin, and how the circular reasoning around $\Gamma_t=0$ is resolved rigorously?
>
> The key idea to remove the dependence on the parameter $\rho$ is to avoid the primal-dual approach. Indeed, in primal-dual methods, the regret guarantees of the primal algorithm and the dual one scale as the Lagrangian variables range, which is set at $1/\rho$, in order to compensate reward and constraints satisfaction. Thus, the regret and violation scale accordingly, leading to vacuous bounds when $\rho$ is small. The main alternative to primal-dual methods is to optimize the rewards over an estimated safe decision space that concentrates to the true one during the learning dynamic. In such a case, **no "compensation" is needed**, since the reward optimization is performed freely in the per estimated safe decision space. The main problem remains how to build the estimated decision space, which is easy in the stochastic setting, where standard optimism is sufficient. Differently, for CMDPs with adversarial constraints, a straightforward approach to build the aforementioned decision space is not known. Remarkably, we show how, by properly associating weights to the past samples, it is possible to build constraint estimators that "moves over time" when the constraints are adversarial, thus encompassing the adversariality of the environment.
>
> Since it is a crucial aspect, please let us know if further discussion is necessary.
>
> We refer to the next answer for the more technical discussion on $\Gamma_{t,i}$.
>
> > Can you formally show that $\Gamma_{t,i}$ remains small (or zero) in the stochastic case and stabilizes in the adversarial case without knowing the regime a priori?
>
> We thank the Reviewer for the opportunity to clarify these aspects. As concerns the first result, that is, $\Gamma_{t,i}=0$ in the stochastic case, this is formalized in the proof of Lemma 4.1. Intuitively, $\Gamma_{t,i}=0$ until the $i$-th constraint violation remain bounded by $\widetilde{\mathcal{O}}(\sqrt{T})$. Additionally, when $\Gamma_{t,i}=0$, we have that the constraints estimator $\widehat{g}\_{t,i}$ is exactly the empirical mean of the past constraints observed as feedback. Finally, noticing that, in stochastic settings, employing an optimistic approach on the constraints based on the empirical mean $\widehat{g}_{t,i}$ and the Hoeffding bonus term $b_t$ is sufficient to keep the violation sublinear, we have the final result.
>
> As concerns the second result, this is formalized in Theorem B.7. Intuitively, when the violation starts to increase, the weights are set so that the samples are not weighted uniformly, behaving as a sliding window on the decision space.
>
> > Since $\rho$ can be arbitrarily small, doesn’t this make $\alpha$-regret potentially vacuous? Can you show any setting where $\alpha$ is meaningfully bounded away from zero?
>
> The Reviewer is correct when stating that the $\alpha$-regret can be potentially vacuous for hard settings, that is, when the strictly feasible solution is not largely safe. Nonetheless, it is standard in the constrained online learning literature to focus on the $\alpha$-regret definition when the constraints are adversarial (e.g., [1] for the MAB case, [2] for the CMDP case, and [3] for the online allocation problem). This is due to the well-known impossibility result of learning with adversarial constraints, namely, it is provably not possible to obtain both sublinear regret and sublinear violation when the constraints are adversarial. As a final remark, we underline that it is provably not possible to attain sublinear violation and sublinear $\alpha$-regret with a better notion of $\alpha$ (see Theorem 4.1 of [9]).
>
> > What is the per-episode computational complexity, and how is $u_t$ computed in your experiments.
>
> The computational complexity of our algorithm mainly lies in the convex projection in the estimated safe set of occupancy measures, which can be approximated arbitrarily well in polynomial time. The upper occupancy bound can be computed algorithmically with a computational complexity of order $\mathcal{O}(|X|^3|A|^2)$ (see [4] for additional details).
>
> We remark that solving convex programs and/or computing upper bounds on occupancy measure are standard even in simpler adversarial unconstrained MDPs (e.g., [4], [5], [6], [7]).
>
> > On the experiments.
>
> We refer to the Appendix for the experiments in the adversarial setting. This choice was primarily motivated by space constraints and the fact that best-of-both-worlds algorithms tend to perform worse in stochastic setting, where benchmarks tailored only for the stochastic regime can be used. We commit to using the additional page in the final version to include the results for the adversarial setting, too.

---

> ### Author Response · Authors · 2025-11-14
>
> > I believe the claim of being ``best-of-both-worlds'' is overstated. The algorithm indeed shares one form across regimes, but its guarantees remain conservative (\emph{$\alpha$-regret}) whenever constraints are adversarial, even if they are benign.
>
> We kindly disagree with the Reviewer on this point. Our definition of "best-of-both-worlds" is commonly accepted in the literature (e.g., [1],[2],[8],[9]). In the online learning with unknown constraints literature, the best-of-both-worlds term refers to algorithms capable of obtaining optimal regret and violation bounds when the constraints may be either stochastic or fully adversarial. In the stochastic regime, our regret and violation bounds are provably tight in $T$ and do not require **any** Slater condition. This is a strong improvement with respect to the state-of-the-art result in [2], where the bounds depend on Slater's parameter $\rho$. The same reasoning holds for the violation in the adversarial setting. Finally, as previously mentioned, the $\alpha$-regret is considered the way to go in the literature. Additionally, our $\alpha$-regret bound does not have the $1/\rho$ dependency as in [1,2]. Moreover, the $\alpha$ factor is used to approximate the unconstrained optimum and not the constrained one, which is an improvement on the state-of-the-art best-of-both-worlds bounds in CMDPs.
>
> Since all these aspects are crucial, please let us know if further discussion is needed. We hope to have addressed the Reviewer’s concerns. In that case, we would be grateful if the Reviewer could reconsider their score accordingly.
>
> [1] "A Unifying Framework for Online Optimization with Long-Term Constraints", NeurIPS 2022
>
> [2] "Policy Optimization for CMDPs with Bandit Feedback: Learning Stochastic and Adversarial Constraints", ICML 2025
>
> [3] "The Best of Many Worlds: Dual Mirror Descent for Online Allocation Problems", Operations Research
>
> [4] "Learning Adversarial MDPs with Bandit Feedback and Unknown Transition", ICML 2020
>
> [5] "Online stochastic shortest path with bandit feedback and unknown transition function", NeurIPS 2019
>
> [6] "Online convex optimization in adversarial Markov decision processes", ICML 2019
>
> [7] "Policy Optimization in Adversarial MDPs: Improved Exploration via Dilated Bonuses", NeurIPS 2021
>
> [8] "Online Learning in CMDPs: Handling Stochastic and Adversarial Constraints", ICML 2024
>
> [9] "No-regret is not enough! bandits with general constraints through adaptive regret minimization", ICML 2025

---

### Official Review · Reviewer_GNMi · 2025-11-01

**Soundness:** 3
**Presentation:** 3
**Contribution:** 3
**Rating:** 4
**Confidence:** 3

**Summary:**

This paper studies online episodic constrained MDPs with adversarial rewards and stochastic or adversarial constraints and proposes an algorithmic framework that achieves desired performance guarantees without Slater's condition being satisfied. In the stochastic constraints regime, the algorithm obtains rate-optimal $\tilde O(\sqrt{T})$ regret and constraint violation bounds without Slater's condition. In the adversarial constraints setting, they take the notion of $\alpha$-regret and prove that the algorithm achieves an $\alpha$-regret bound of $\tilde O(\sqrt{T})$. The algorithm avoids a primal-dual Lagrangian update rule, which often leads to a dependence on Slater's constant. Instead, they take feasible sets that encourage optimistically satisfying constraints. Based on this innovative technique, the paper ends up achieving Slater constant-free bounds.

**Strengths:**

- The theoretical contribution is that it obtains the first performance bounds for learning online CMDPs that do not require satisfying Slater's condition.
- The algorithm design is novel. It is natural to take an OMD-style policy update for learning online MDPs. The novelty, however, is that instead of penalizing the constraint via Lagrangian multipliers, they impose an optimistic constraint when taking an approximate feasible set in each episode. This leads to the Slater constant-free regret and constraint violation bounds.

**Weaknesses:**

The approach of taking optimistic constraints when constructing feasible sets reminds me of LP-based approaches for learning CMDPs. In fact, Efroni et al. (https://arxiv.org/pdf/2003.02189) have already come up with an algorithm for the stochastic setting that achieves (strong) regret and (strong) constraint violation bounds without Slater's condition. It seems that for the stochastic setting, the bounds due to Efroni et al. are tighter in terms of the number of states. It would be more exciting if one could derive an algorithm that achieves a zero constraint violation bound and a regret bound that does not depend on Slater's constant (See Bura et al. https://proceedings.neurips.cc/paper_files/paper/2022/hash/076a93fd42aa85f5ccee921a01d77dd5-Abstract-Conference.html).

**Questions:**

- Can the algorithm be modified to achieve a zero constraint violation bound and a regret bound that does not depend on Slater's constant?

---

> ### Author Response · Authors · 2025-11-14
>
> We thank the Reviewer for the effort in evaluating our work.
>
> > The approach of taking optimistic constraints when constructing feasible sets reminds me of LP-based approaches for learning CMDPs. In fact, Efroni et al. have already come up with an algorithm for the stochastic setting that achieves (strong) regret and (strong) constraint violation bounds without Slater's condition. It seems that for the stochastic setting, the bounds due to Efroni et al. are tighter in terms of the number of states.
>
> The Reviewer is correct in stating that a UCB approach on the constraints was originally introduced by [Efroni et al., 2020], in their Algorithm $1$. Nonetheless, we remark that their approach does **not** work when the constraints are adversarial, thus not weakening the results presented in our work. As concerns the dependence on the number of states, notice that the bounds in [Efroni et al., 2020] have an additional dependency on the quantity $\mathcal{N}$ that can be potentially as large as the set of states, thus collapsing to our violation bound.
>
> > It would be more exciting if one could derive an algorithm that achieves a zero constraint violation bound and a regret bound that does not depend on Slater's constant (See Bura et al.). Can the algorithm be modified to achieve a zero constraint violation bound and a regret bound that does not depend on Slater's constant?
>
> Without assuming Slater's condition, it is provably not possible to achieve $0$ constraints violations (see Theorem 3 of [1], for the simpler bandit case). Similarly, even assuming Slater's condition, it is provably not possible to achieve a zero constraint violation bound and a regret bound that does not depend on Slater's constant (see Theorem 6.6 of [2]).
>
> We hope to have addressed the Reviewer’s concerns. In that case, we would be grateful if the Reviewer could reconsider their score accordingly. Please let us know if further discussion is needed.
>
>
> [1] "Safe Learning in Tree-Form Sequential Decision Making: Handling Hard and Soft Constraints", ICML 2022
>
> [2] "Learning Adversarial MDPs with Stochastic Hard Constraints", ICML 2025

---

> > ### Comment · Reviewer_GNMi · 2025-11-26
> >
> > Thank you for the response.
> >
> > It is correct that Algorithm 1 of [Efroni et al.](https://arxiv.org/pdf/2003.02189) may not work when the constraints are adversarial. However, as stated in the review, for the stochastic setting, their regret bound is of order $\widetilde{\mathcal{O}}(S\sqrt{AT})$ (see also [Bura et al.](https://proceedings.neurips.cc/paper_files/paper/2022/hash/076a93fd42aa85f5ccee921a01d77dd5-Abstract-Conference.html)), which is better than the regret bound of order $\widetilde{\mathcal{O}}(S^2\sqrt{AT})$ established in this paper.
> >
> > I appreciate the two references in response to my question.
> >
> > As I am leaning toward acceptance of this paper, I have raised my score accordingly.

---

### Official Review · Reviewer_Qs19 · 2025-11-05

**Soundness:** 4
**Presentation:** 3
**Contribution:** 3
**Rating:** 6
**Confidence:** 3

**Summary:**

The paper studies online episodic CMDP and improve on the best-of-both worlds algorithm achieving better regret and violation guarantees $\tilde{\mathcal{O}}(\sqrt{T})$ for both stochastic and adversarial constraint and only adversarial reward which is a more general case. The paper also introduces the WC-OPS algorithm, which achieves sublinear regret theoretically. Finally, it demonstrates a few experiments.

**Strengths:**

1.  The paper is theoretically sound.
2. The perfect bounds are certainly helpful
3. The algorithm has merits and definitely the paper paves the direction of new research

**Weaknesses:**

1. Many typos. Listing a few
      a) Page 2 footnote 2 the simplex dimension is incorrect. Either it should be n dimensional or line 3 under section 2.1 the definition of probability transition function (|X|+1) and anywhere else simplex is used, it has accounted $\Delta_{n}$ to be n dimensional and not n-1 dimensional.
     b)  $X_{k}, X_{k+1}$ not defined
     c) Under the section 3.1 the definition of $M_{t}(x^{'}|x,a)$ uses $t$ as subscript which should be $\tau$. Later in the same paragraph the inidicator functions were mistyped again.
    d) Section 3.3 definition of $\epsilon_{t}(x,a)$, denominator is max{.,.} and not $max 1,N_{t}...$
These typos make it difficult to read

2. Under section 2.1 the conditions for occupancy is dependent on initial state distribution but not mentioned.
3. A little more clarity on the weights estimation would be good. This is good, but not very practical.
4. Abrupt end of the main paper. It seems the paper could not have a proper closure due to page restriction. I suggest moving some details to the appendix.

**Questions:**

1. The experimental setup seems very unrealistic. Just curious whether the algorithms can be projected to some benchmarks or not.

2. Curious about the effect of bonus terms on $\gamma$ and the results (empirical).

3. Decising the $\alpha$ term seem very difficult or solving an optimization problem might be difficult or time taking. Curious how thw authors decide on the same during their experiments.

---

> ### Author Response · Authors · 2025-11-14
>
> We thank the Reviewer for the positive evaluation of our work.
>
> > On the presentation.
>
> We thank the Reviewer for pointing out the typos and suggesting possible improvements in the presentation. We will surely include them in the final version of the paper.
>
> > The experimental setup seems very unrealistic. Just curious whether the algorithms can be projected to some benchmarks or not.
>
>  As is standard in the online learning and regret minimization literature, we compare against state-of-the-art algorithms. In the main paper, we focus on algorithms specifically tailored for the stochastic setting, as best-of-both-worlds methods typically perform worse in this case. In the Appendix, we also include experiments under the adversarial settings.
>
> > Curious about the effect of bonus terms on $\gamma$ and the results (empirical).
>
> We are not entirely sure whether we have properly understood the focus of the question. If the Reviewer is referring to the hyperparameter $\gamma$, it is generally set to the same value as the learning rate $\eta$. Nonetheless, small (constant) variations of $\gamma$ do not significantly impact the performance.
>
> If the question concerns the adaptive term $\Gamma_{t,i}$ and its relation to the bonus terms $b_t$, we notice that in the stochastic setting $b_t$ is crucial to ensure optimism in the definition of the decision space, while $\Gamma_{t,i}=0$. In contrast, in the adversarial setting, $\Gamma_{t,i}$ becomes active and is the main mechanism shaping the decision space, while $b_t$ is not required.
>
> > Decising the $\alpha$ term seem very difficult or solving an optimization problem might be difficult or time taking. Curious how the authors decide on the same during their experiments.
>
> In our experiments, the parameter $\alpha$ is not manually tuned: it derives from the problem-specific quantity $\rho$ ($\alpha = \rho/(1+\rho)$) (Section 2.3). In practice, $\rho$ is computed taking the worst constraints state-action value of the safest occupancy measure.

---

> > ### Comment · Reviewer_Qs19 · 2025-11-28
> >
> > I thank the authors for clearing the doubts and I lean towards acceptance of the paper.

---

### Meta-Review · Area_Chair_FncT · 2026-01-05

**Summary:**

This work addresses CMDPs under stochastic and adversarial conditions. Its principal technical merit lies in imposing constraints for optimization as an alternative to Lagrange multiplier balancing. Reviewer opinions were split, with a major point of contention being the theoretical analysis for the adversarial setting. Specifically, the adopted regret criterion is considered weak, which compromises the significance of the results in that regime. As the paper stakes its primary contribution on being a "best-of-both-worlds" framework, this constitutes an overclaim that is not adequately supported. Consequently, my recommendation is to reject.

**Reviewer Concerns:**

Concerns addressed by the rebuttal:
1) The necessary assumptions and conditions for the algorithm's validity are well clarified;
2) Computational cost of the optimization problem;
3) Comparison with related works (e.g., Efroni et. al., 2020);

Concerns remained after the response:
1) Bad regret criteria in the adversarial case;
2) The selection of a crucial parameter $\rho$ is not clear. As a result,  the  "best-of-both-worlds" claim appears overstated;
3) Limited numerical experiments.

**Reviewer Scores:**

Reviewer GNMi claimed he would increase his score. I expect the rest of the reviewers would have kept their scores.

---

### Decision · Program_Chairs · 2026-01-26

Reject